# Live cell dynamics of production, explosive release and killing activity of phage tail-like weapons for *Pseudomonas* kin exclusion

Jordan Vacheron [1,2✉], Clara Margot Heiman [1,2] & Christoph Keel [1✉]

Interference competition among bacteria requires a highly specialized, narrow-spectrum weaponry when targeting closely-related competitors while sparing individuals from the same clonal population. Here we investigated mechanisms by which environmentally important *Pseudomonas* bacteria with plant-beneficial activity perform kin interference competition. We show that killing between phylogenetically closely-related strains involves contractile phage tail-like devices called R-tailocins that puncture target cell membranes. Using live-cell imaging, we evidence that R-tailocins are produced at the cell center, transported to the cell poles and ejected by explosive cell lysis. This enables their dispersal over several tens of micrometers to reach targeted cells. We visualize R-tailocin-mediated competition dynamics between closely-related *Pseudomonas* strains at the single-cell level, both in non-induced condition and upon artificial induction. We document the fatal impact of cellular self-sacrifice coupled to deployment of phage tail-like weaponry in the microenvironment of kin bacterial competitors, emphasizing the necessity for microscale assessment of microbial competitions.

[1] Department of Fundamental Microbiology, University of Lausanne, CH-1015 Lausanne, Switzerland. [2] These two authors contributed equally: Jordan Vacheron, Clara Margot Heiman. ✉email: jordan.vacheron@unil.ch; christoph.keel@unil.ch

Pseudomonads colonize various terrestrial and aquatic environments and associate with diverse hosts including plants, invertebrates and humans[1]. These bacteria have evolved mechanisms to cope with different stresses inherent to living in these environments. *Inter alia*, they need to outcompete other contenders to durably establish their presence in the colonized environment. Strains belonging to the *Pseudomonas protegens* (Pp) or *Pseudomonas chlororaphis* (Pc) subgroups within the *Pseudomonas fluorescens* species complex[2] typically are highly competitive root colonizers that, however, are also able to establish themselves in contrasting ecological niches, notably in insects[3,4]. These bacteria are widely studied for their potential use as agricultural inoculants for plant growth promotion, defense priming and protection against pathogens and pest insects[5–7]. A part of their competitiveness is due to the production of an array of secondary metabolites with broad antimicrobial activities against bacteria, fungal and protist competitors that would fight for common resources[8,9].

As well as competing against distantly-related organisms, pseudomonads are in competition with kin bacteria, i.e., phylogenetically close relatives, which we consider as strains belonging to the same subgroup in this study. Therefore, they also produce narrow-spectrum toxins, called bacteriocins that typically target close relatives[10]. These compounds vary greatly in structure and in activity profiles owing to their specificity[10]. There is some evidence that bacteriocins might shape natural communities of environmental *Pseudomonas* but the extent of their impact has yet to be defined[11,12].

Bacteriophages and phage-like particles also exhibit specific activity spectra and have emerged as potent specific weapons against competitors[13,14]. Indeed, bacteriophages were shown to influence microbiomes through their lysogenic and lytic cycles[15,16]. For instance, phages may impact species diversity by multiple mechanisms such as reducing the number of highly competitive bacteria ("killing the winner"), releasing nutrients and DNA into the environment and immunizing bacteria against similar viral relatives or equip them with new traits[15,17–19].

When bacteriophages integrate their genome into bacteria, it may give rise to complex structures such as phage tail-like particles that can be used against other competitors[20–22]. One of these structures was first discovered in *Pseudomonas aeruginosa* and was termed pyocin by analogy with colicins[23]. Similar structures were found in other bacterial species and broadly designated as tailocins[22,24] or phage tail-like bacteriocins[25]. There are two main types of tailocins, R-type and F-type, typified by the *P. aeruginosa* R-pyocins and F-pyocins, respectively[10,23,25]. F-tailocins are flexible and non-contractile rod-like structures, whereas R-tailocins are large, rigid and contractile tail-like structures[10,23,24]. Although F-type and R-type tailocins are thought to be evolutionarily related to phages from the families *Siphoviridae* and *Myoviridae*, respectively, these structures are not considered degenerated phages but evolved powerful bacterial weapons[24,26]. R-tailocins are structurally related to a large family of bacterial contractile injection systems, which comprises the type VI secretion system (T6SS) and phage tail-like insecticidal particles[21,27].

The genomic loci encoding R-tailocins in pseudomonads specify structural components as well as the regulators and lysis cassette permitting a timely production and release of the particles[25,28–31]. Conversely, R-tailocin clusters no longer comprise the genes coding for the capsid that encapsulates the phage genomic material and makes a normal phage self-replicative. Analogous to the contractile tail of certain bacteriophages, R-tailocins are composed out of a sheath and a hollow tube forming a long helicoidal hexameric structure that is attached to a baseplate exhibiting multiple tail fibers that appear to direct specificity and permit adsorption to the cell surface of the targeted cell[23–25,27,32,33]. The contractile tail is thought to cause a dissipation of the membrane potential, leading to the death of the targeted cell[25,32,34]. The conditions triggering tailocin production naturally are basically unknown. In the laboratory, tailocin assembly can be induced by DNA damage stress provoking the SOS response[10,23,25]. Tailocin release then appears to be mediated by components of the lysis cassette (holins, lysins and spanin complexes) that degrade the cell wall of the triggered bacterium[10,25,30].

Environmental *Pseudomonas* were recently shown to harbor gene clusters encoding R-tailocins[28–31]. However, there is still very limited knowledge about the activity profiles, the involvement in competition or the production dynamic of these complex phage tail-like particles in environmental pseudomonads. Therefore, in this study, we assessed the diversity of R-tailocins and other phage particles belonging to strains of the Pp and Pc subgroups and related it to their activity spectra within these phylogenetically very closely-related environmental pseudomonads. Then we focused on the type strain of the Pp subgroup, namely CHA0, that is a well-studied model organism for environmental *Pseudomonas*. For the first time to our knowledge, we visualized the production dynamic of R-tailocins in a bacterial cell and their ejection upon explosive cell lysis. By live cell imaging, we documented competitions involving R-tailocins between two closely-related strains at single-cell level, both in non-induced condition and upon artificial induction, evidencing that this phage tail-related weaponry can play a decisive role in the micro-environment of kin competitors.

## Results

**Environmental pseudomonads have a large diversity in viral particles that exhibit phylogenetic-specific activities.** To better understand the role of viral particles as weapons in competition between kin bacteria, we selected phylogenetically closely-related environmental *Pseudomonas* of the Pp and Pc subgroups that encompass the bacterial model *Pseudomonas protegens* type strain CHA0 (Supplementary Table 1). We first looked at the distribution of viral particle encoding loci in the genomes of this set of *Pseudomonas* strains using the prediction tool PHASTER (PHAge Search Tool Enhanced Release[35]), manually verifying each region to classify them as either incomplete or complete prophages. None of the *Pseudomonas* genomes selected is free of viral particle loci (Fig. 1). Two viral families were identified: *Siphoviridae* and *Myoviridae*. PHASTER also classified as intact prophages another set of gene clusters that are incomplete, as they no longer contain the genes encoding the synthesis of the viral capsid. These regions correspond to gene clusters encoding phage tail-related structures designated as R-tailocins and are present in all *Pseudomonas* genomes used in this study (Fig. 1). Interestingly, there is no correlation with the number or the classification of encoded viral particles and the phylogenetic position of a strain.

Next, we extracted the viral particles from mitomycin C (MMC) induced cultures of the selected *Pseudomonas* strains and cross-tested them on the same strains to determine their activity spectra. We found that strains were overall more sensitive to the viral particles extracted from phylogenetically closely-related relatives (i.e., from the same subgroup, Pp or Pc) and even to the viral particles extracted from the same species (Fig. 1). Additionally, some *Pseudomonas* strains, specifically AU13582, AU11706 and AU20219, had no targets within the tested Pp and Pc collection (even though they harbor viral particle loci in their genomes) (Fig. 1). This absence of effect may also be due to the non-production of viral particles following induction. Indeed,

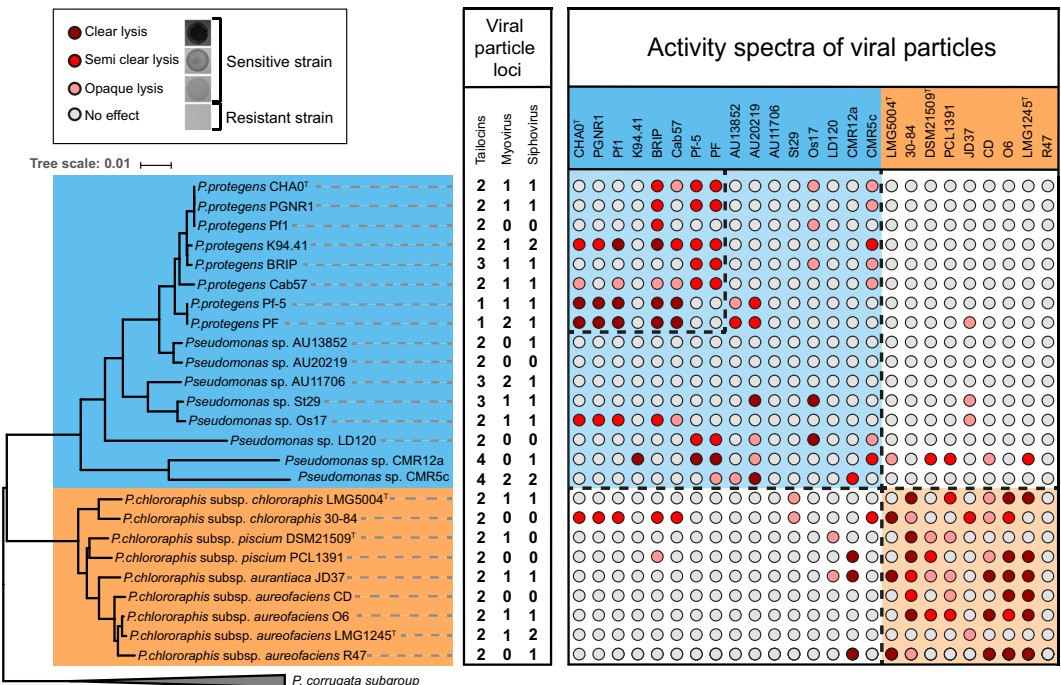

**Fig. 1 Intraspecific sensitivity of *Pseudomonas protegens* (Pp) and *Pseudomonas chlororaphis* (Pc) subgroup strains when confronted to their viral particles.** The viral particle loci inventory was generated using PHASTER[35] and manually inspected to discriminate tailocin gene clusters from prophages by checking for genes encoding capsid-related proteins. The prophages were classified as either siphoviruses or myoviruses. No other phage family was detected. Soft agar cultures of the *Pseudomonas* strains (columns) were challenged with viral particles extracted from cultures of these same strains (rows). Strains producing clear, semi-clear or opaque lysis zones were scored sensitive to the viral particles, while strains producing no lysis zone were considered resistant. The boxes with the dashed lines in the table delimits the *sensu-stricto P. protegens* species, the Pp subgroup (blue) and the Pc subgroup (orange).

some prophages can be maintained as defective prophages[20]. *Pseudomonas* sp. AU11706 and *Pseudomonas chlororaphis* R47 are the only strains that were resistant to all the viral particles extracted from the strains tested in this study (Fig. 1). *P. chlororaphis* O6 appeared to be sensitive to its own viral particles (Fig. 1). This self-sensitivity reaction could be a cause of the reinfection by one of the two prophages contained in this bacterial strain.

Together these results support the hypothesis that some viral particles are produced to target particularly phylogenetically closely-related strains.

**Viral particle dissection unveils individual species and subgroup target specialization.** When we focused on the model CHA0, PHASTER found three prophage-like sequences (Fig. 1). Two of these regions encode complete phages belonging to the *Myoviridae* and *Siphoviridae* families, respectively. Conversely, the third region lacks the genes specifying capsid formation and was identified as the gene cluster encoding the production of two distinct phage tail-like particles that we termed tailocin #1 and tailocin #2 of CHA0 (see below). To determine the specific activity spectrum of each viral particle of CHA0, we deconstructed them by deletion of their coding regions (Supplementary Tables 2, 3, 4) and tested the individual and combined effects of the extracted viral particles on the *Pseudomonas* strains of this study. A total of six strains were impacted by at least one of the viral particles of CHA0 (Fig. 2a). Affected strains all belong to the Pp subgroup (i.e. BRIP, Cab57, Pf-5, PF, Os17 and CMR5c), amongst which four belong to the *P. protegens sensu stricto* species (Fig. 1), alongside type strain CHA0 (as they have more than 97% of genomic identity). None of the Pc subgroup strains were affected by the viral particles of CHA0 (Fig. 2a). Moreover,

there is a specialization of each viral particle in accordance with the *Pseudomonas* phylogeny. Tailocin #1 impacted strains from the broader Pp subgroup (i.e. *P. protegens* PF, Pf-5, Os17 and CMR5c), while the strains affected by the siphovirus are part of the same species (i.e., *P. protegens* BRIP, Cab57 and Pf-5) (Fig. 2a). Only *P. protegens* Pf-5 was sensitive to both tailocin #1 and the siphovirus. The sensitivity of those strains to either tailocin #1 and the siphovirus were similar according to serial dilutions performed (Supplementary Data 1). Tailocin #2 and the myovirus of CHA0 did not lyse any of the *Pseudomonas* strains tested (Fig. 2a).

Using the same mutagenesis approach, we were able to produce and visualize independently three of the predicted viral particles of CHA0, i.e. tailocin #1, tailocin #2 and the siphovirus, by transmission electron microscopy (TEM) (Fig. 2b). We observed the two R-tailocins of CHA0 in the extended (sheath uncontracted) and contracted (sheath contracted, and tube expulsed) forms. Furthermore, the size differed between both R-tailocins as tailocin #1 was longer than tailocin #2 (Fig. 2b, Supplementary Table 5). However, we were not able to visualize the predicted myovirus that we suspect to be a cryptic prophage. This hypothesis is supported by the growth curve of the mutant with only the sequence of the myovirus (ΔtailclusterΔsiph) as the cells behave in a similar manner as the mutant lacking all regions encoding viral particles (ΔtailclusterΔsiphΔmyo) in MMC inducing conditions. In fact, in the absence of MMC induction, the mutations leading to the deletion of the different viral particles of CHA0 did not affect bacterial growth (Supplementary Fig. 1a). When MMC was added, only the mutant no longer producing viral particles (ΔtailclusterΔsiphΔmyo) as well as the mutant harboring only the myovirus sequence (ΔtailclusterΔsiph) continued to elongate and did not lyse due to the removal of their lytic systems (Supplementary Fig. 1b, 1c, 1d). By contrast, a

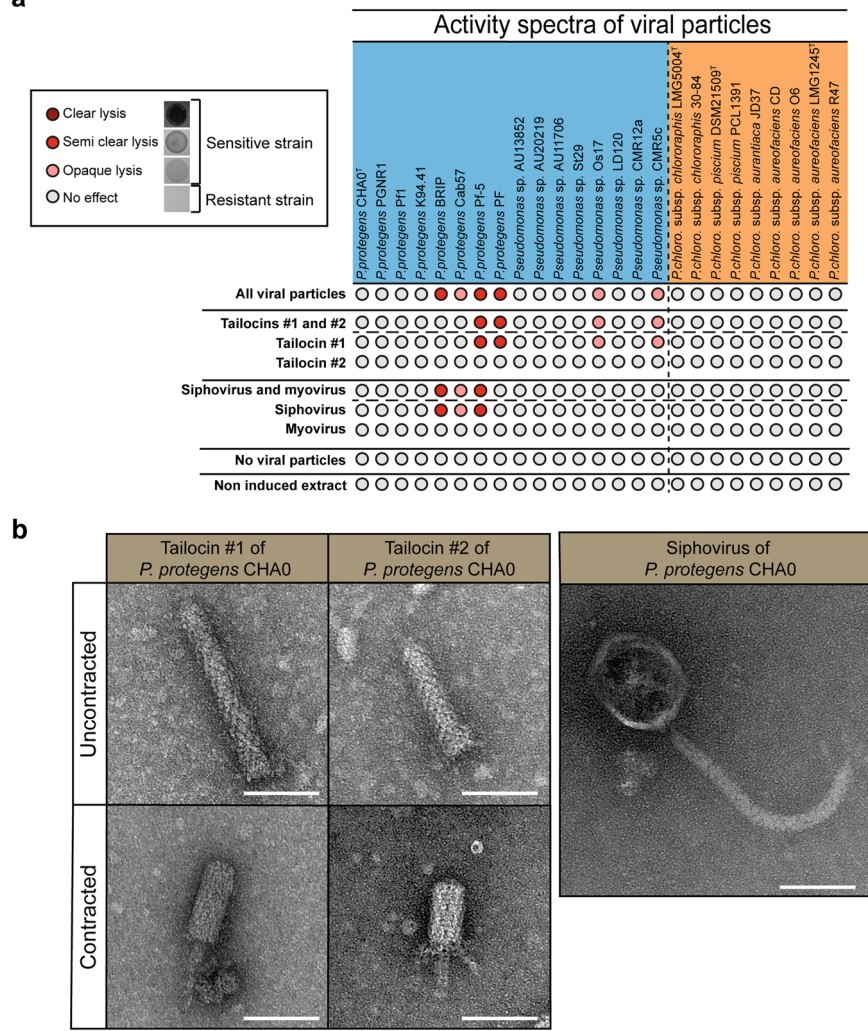

**Fig. 2 The viral particles of *Pseudomonas protegens* CHA0 exhibit different activity spectra towards strains belonging to the *Pseudomonas protegens* (Pp) subgroup and the *Pseudomonas chlororaphis* (Pc) subgroup. a** Pp (blue) and Pc (orange) strains were challenged with viral particles extracted from the following mutants that were induced or not by 3 µg mL$^{-1}$ of mitomycin C: Tailocins #1 and #2, ΔmyoΔsiph, CHA5299; Tailocin #1, Δtail2ΔmyoΔsiph, CHA5302; Tailocin #2, Δtail1ΔmyoΔsiph CHA5301; Siphovirus and myovirus, Δtailcluster, CHA5285; Siphovirus, ΔtailclusterΔmyo CHA5289; Myovirus, ΔtailclusterΔsiph CHA5292 (Supplementary Table 2). Serial dilutions of the viral particles were performed and used to challenge all *Pseudomonas* strains. The results are available in the Supplementary Data 1. Tested *Pseudomonas* strains are organized in a phylogenetic manner as in Fig. 1. **b** Electron microscopy photographs of viral particles produced by *P. protegens* CHA0. Tailocin #1 and #2 are shown either in the non-contracted or contracted forms. The siphovirus possesses a characteristic long, flexible and non-contractile tail. The myovirus of CHA0 was not detected upon the tested induction conditions. Individual viral particles were prepared from the above described CHA0 derivatives. The scale bar corresponds to 60 nm.

mutant lacking all viral particles but still harboring the lytic enzymes encoded by the R-tailocin gene cluster (Δtail1Δtail2Δ-myoΔsiph) lysed in presence of MMC (Supplementary Fig. 1b).

This demonstrates that the viral particles of CHA0 are produced following MMC induction and affect strains from the same subgroup and highly impact strains belonging to the same species.

**Tailocin diversity mirrors the phylogenetic distance between pseudomonads**. As tailocins display a high specificity, we then centered our focus on the genomic region encoding these viral particles. The R-tailocin genomic region of the model CHA0 is flanked by the genes *mutS* and *cinA/recA*, respectively. It is composed out of 36 phage-like genes and encodes the structural components to assemble the two distinct R-tailocins (tailocin #1 and tailocin #2) identifiable by TEM (Fig. 2b), including their baseplates, tail spikes, tubes, sheaths, tail fibers, and putative

assembly chaperones (Fig. 3a, Supplementary Table 4). Each CHA0 R-tailocin is predicted to have its own tail tape measure protein (Fig. 3a) that by analogy with related proteins[36,37] is thought to scaffold the respective tail tubes during their assembly. Interestingly, the coding sequence of the tail tape measure protein for tailocin #1 is markedly longer than the one for tailocin #2, which explains the different lengths of the two particles (Supplementary Table 5). The R-tailocin cluster of CHA0 is also equipped with a set of genes encoding a holin, two endolysins and a spanin complex (Fig. 3a, Supplementary Table 4), i.e. all components known to be required for the lysis of the bacterial cell wall and the release of the phage-like particles[38]. The organization of the genes specifying these structural components is different for the two R-tailocins indicating that they have most probably evolved from two independent prophage infections.

In the selected Pp and Pc genomes, we found that the genomic region situated between *mutS* and *cinA/recA* is a hotspot for the integration of tailocin clusters (Fig. 3b), as it was already

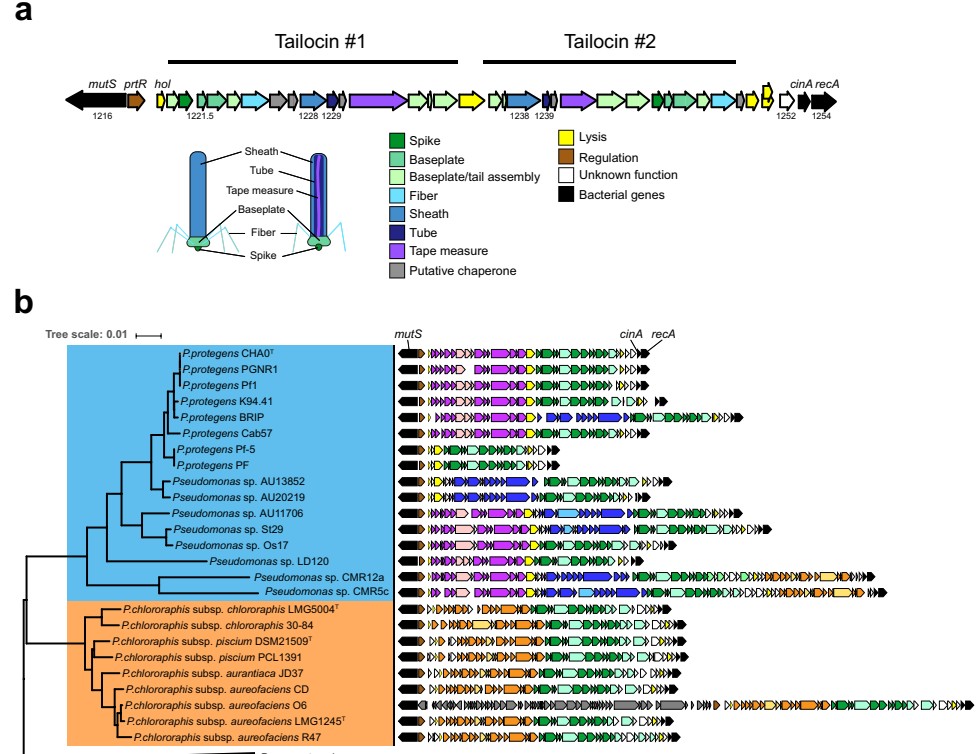

**Fig. 3 Architecture of the R-tailocin gene cluster of *Pseudomonas protegens* CHA0 and diversity of tailocin clusters among strains of the *Pseudomonas protegens* (Pp) subgroup and *Pseudomonas chlororaphis* (Pc) subgroup. a** The R-tailocin gene cluster of *P. protegens* CHA0 is composed out of 36 phage-related genes that encode two distinct R-tailocin complexes (tailocin #1, tailocin #2). A detailed description is given in the Supplementary Table 4. **b** The tailocin loci are situated between the bacterial genes *mutS* and *cinA/recA* (black arrows) of Pp (blue) and Pc (orange) strains. Groups of homologous genes are indicated using the same color. The identity levels between these latter are identified using different color intensities: dark colors describe sequences sharing more than 70% of nucleotide identity on at least 70% of the total gene length while light colors represent sequences sharing less than 70% of identity on at least 70% of the total gene length. Four distinct types of tailocin gene clusters were identified: purple, green and orange colored tailocin clusters were identified as encoding R-tailocins while the blue colored cluster encodes an F-tailocin. One entire prophage (gray color) was detected in the tailocin locus of *Pseudomonas chlororaphis* subsp. *aureofaciens* O6. Genes encoding the regulator PrtR (brown), lysis-related proteins (yellow) and hypothetical proteins (white) are indicated.

described for some pseudomonads[28–31]. The analysis of the synteny of the various genomic clusters identified loci encoding four different viral particle groups, i.e., three distinct R-tailocins and one F-tailocin (Fig. 3b). One of the R-tailocin loci is conserved amongst all the genomes selected and is conserved with tailocin #2 of CHA0 (green-colored gene clusters), while the other two R-tailocin loci are separated between the Pp, which corresponds to tailocin #1 of CHA0 (purple-colored gene clusters), and Pc subgroups (orange-colored gene clusters) (Fig. 3b). The F-tailocin locus is only present in the genome of some strains belonging to the Pp subgroup (blue-colored gene clusters, Fig. 3b). Interestingly, within this hotspot region, the strain *P. chlororaphis* subsp. *aureofaciens* O6 carries a complete prophage in addition to two R-tailocin loci (Fig. 3b). We hypothesize that this is a recent acquisition that may be later domesticated to form a tailocin-type weapon. Although the clusters of the four different types of tailocins are conserved amongst the various genomes, there are differences, notably in the genes encoding the tail fibers (Fig. 3b). This is probably a consequence of the specialization in activity spectra as these structures are involved in host specificity[39,40]. Additionally, there is only a limited similarity amongst the genes located at the extremities of the clusters (Fig. 3b). All the investigated tailocin loci encode a protein similar to PrtR, a regulator of tailocin production in pseudomonads[31], as well as conserved proteins involved in the release of these particles such as holins and lysins (Fig. 3b).

To conclude, tailocin loci are omnipresent in the genomes of the selected *Pseudomonas* strains and, even though they are conserved between the different strains, the significant variations of sequences encoding some structural components, notably tail fibers, may account for the various specificities and divergences in the activity spectra of the different viral particles.

**Visualization of the production dynamic of R-tailocins reveals a specific migratory pattern prior to cell lysis.** Following the analysis of the R-tailocin gene cluster of CHA0, we monitored the cellular production dynamics and the release of both R-tailocins simultaneously by fluorescently labeling the sheath proteins of tailocin #1 and tailocin #2 with mScarlet-I and sfGFP, respectively, and visualizing the cells with time-lapse microscopy following induction with MMC.

During the first 90 min post-induction, cells elongated (Fig. 4a, Supplementary Movie 1). This can be explained by the fact that MMC induces double-strand breaks in the DNA affecting DNA replication and inhibiting cell division[41,42]. Fluorescent R-tailocins started being visible around 90 min post induction (Fig. 4a, Supplementary Movie 1). Interestingly, the R-tailocins of CHA0 appeared to be produced at the center of the cell and then migrated to the cellular poles where they accumulated and formed distinct foci prior to cell lysis (Fig. 4a, b, Supplementary Movies 1, 2).

Following the re-localization of the R-tailocins, the cells lost their longitudinal shape and became spherical (148–154 min)

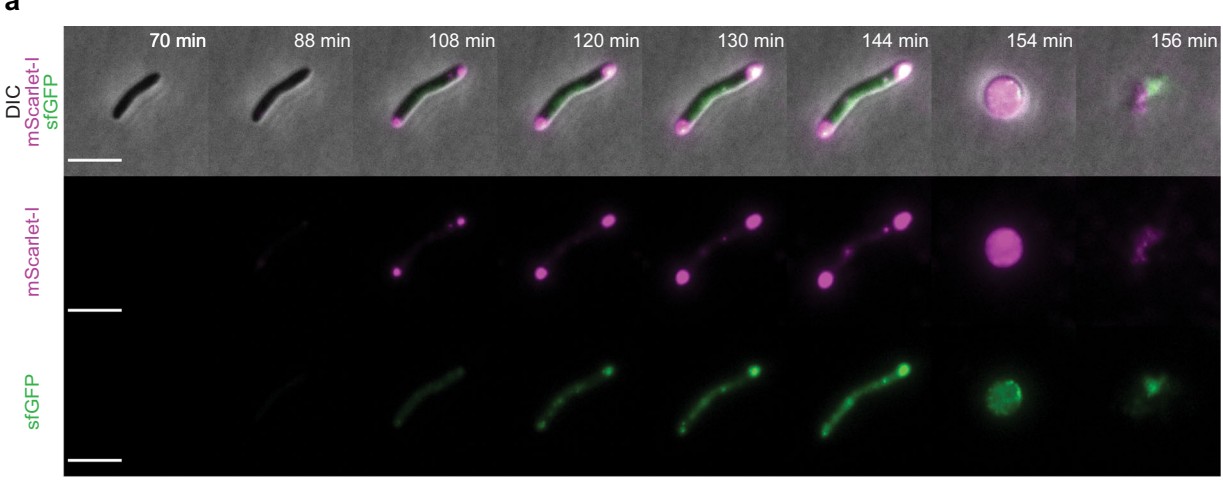

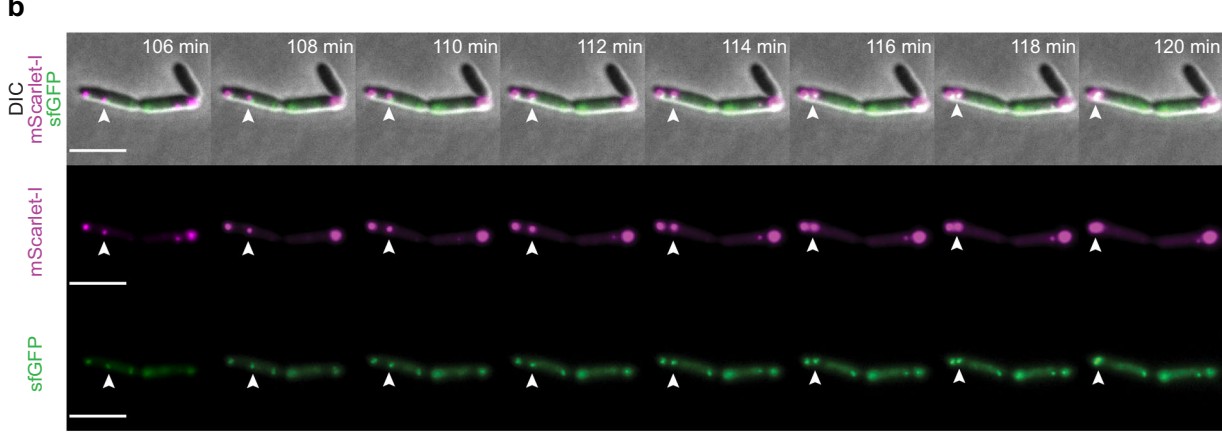

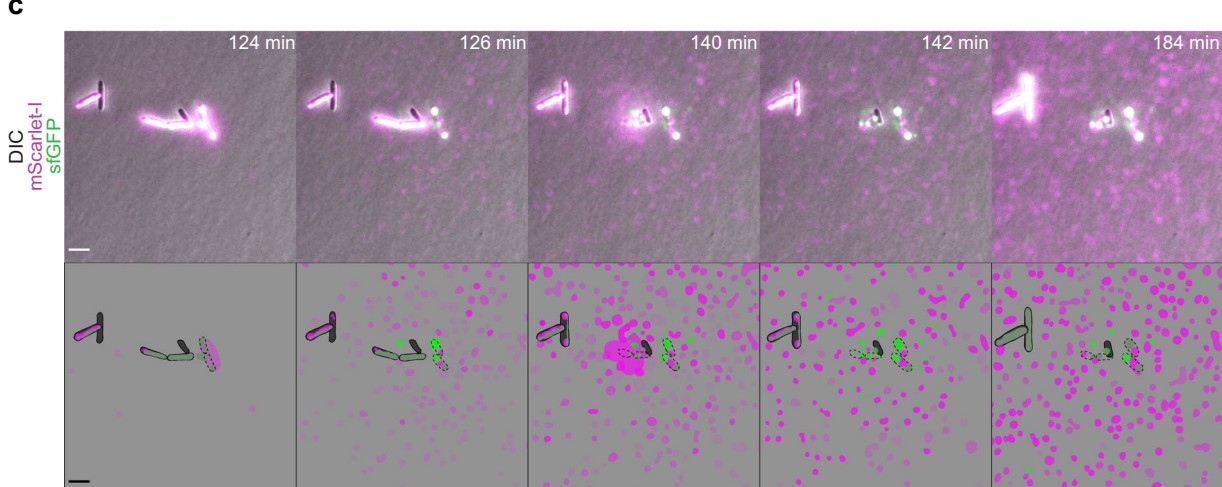

**Fig. 4 Temporal and spatial formation and release of tailocins #1 and #2 in _Pseudomonas protegens_ CHA0 cells following induction with mitomycin C.** Panels **a, b, c** show the different examples of the production dynamic of the two R-tailocins. **a** and **b** R-tailocins are produced at the center of the cell and migrate to cell poles. Following this migration, the cell undergoes a two-step lysis: firstly by forming a spheroblast and then completely lysing. **c** When the cell lyses, R-tailocins are thrusted into the medium. The proteins making up the sheaths of the tailocins #1 and #2 were tagged with the fluorescent proteins mScarlet-I (magenta) and sfGFP (green), respectively. Cells were monitored post induction using time-lapse microscopy (see Supplementary Movies 1, 2, 3); time post induction is indicated in the upper right corner of panels. In panel **c**, the brightness was artificially enhanced to better visualize the R-tailocins in the medium. The succinct increase in tagged tailocins in the background in particular between time points 142 min and 184 min is due to cells out of the frame also producing tailocins. Cartoons depict the release of R-tailocins at different time points. CHA0 live cells are shown in gray while ghost cells are outlined with a dotted line. Tailocins #1 and #2 are illustrated in magenta and green, respectively. The scale bars represent 5 μm.

before completely lysing (156 min) (Fig. 4a, Supplementary Movie 1). The transition between spherical shape and complete lysis occurred rapidly. Strikingly, when cells completely lysed, there appeared to be a strong ejection force thrusting fluorescent R-tailocins far from their point of origin (several tens of micrometers) (Fig. 4c, Supplementary Movie 3). We observed this extent of tailocin dispersion in all time-lapse experiments performed (more than 10 with at least 10 different positions observed). Although this may be a consequence of explosive cell lysis, it could be a mechanism to target cells that are further away. After cellular lysis, both fluorescently labeled R-tailocins became visible in the background (Fig. 4c, Supplementary Movie 3), demonstrating that once ejected from the producing cells, R-tailocins speckle the medium.

**Visualization of the killing activity of individual R-tailocin in direct competition between two strains.** Once produced and thrusted in the medium, R-tailocins are thought to adsorb to the surface of targeted competing cells[25]. Hence, we wanted to determine if R-tailocins have an effect on the competitiveness of cells during competition towards their kin, both in non-induced condition and upon artificial induction of tailocin production by MMC. Therefore, we selected a strain sensitive to tailocin #1 of CHA0, namely Pf-5 (Fig. 2), and first confirmed its sensitivity to this viral particle in liquid cultures through the addition of purified tailocin #1 (Supplementary Fig. 2c). Second, we observed Pf-5 in direct competition with CHA0 expressing fluorescently labeled R-tailocins (tailocin #1 with mScarlet-I, tailocin #2 with sfGFP) using time-lapse microscopy. As the production of R-tailocins is a rare event without induction (less than 1%, Supplementary Fig. 3, Supplementary Data 2), since their production is a fatal decision for the individual cell, we induced their production in CHA0 with MMC prior to placing the so-primed strain into cell-to-cell competition with Pf-5. CHA0 cells were thoroughly washed 30 min following induction as not to induce Pf-5 cells by the chemical. As observed previously, following the induction of CHA0, fluorescent R-tailocins were produced at the center of the cell and migrated to the cell poles (Fig. 5, Supplementary Movies 4, 5). Following the production of the R-tailocins, CHA0 cells lysed and thrusted the viral particles in the environment where they attacked cells of the competitor Pf-5. The production of R-tailocins and the lysis of different CHA0 cells resulted in the lysis of several Pf-5 cells in their vicinity (Fig. 5, Supplementary Movies 4, 5), providing live cellular support for the idea that these phage tail-like devices are efficient weapons for killing kin competitors. However, we expected that the production of R-tailocins would have a more drastic impact on the competitor as Pf-5 is notably sensitive to preparations of purified tailocin #1. We hypothesized that the presence of the fluorescent tag may have an influence on the conformation of tailocin #1, thus impacting its activity. Indeed, examination of purified mScarlet-I tagged tailocin #1 by TEM revealed that the tagged sheath is more compact and therefore it likely has more difficulty efficiently contracting compared to the native form (Supplementary Fig. 2a, b, Supplementary Table 5). Activity testing against Pf-5 cells in liquid cultures confirmed a loss of efficacy compared to the native form (Supplementary Fig. 2c, d).

Considering this bias, we confirmed the important role of the R-tailocins during live bacterial competition using CHA0 mutants defective for all viral particles except tailocin #1 or tailocin #2, respectively. Importantly, assays were carried out in non-induced conditions, i.e. without prior MMC induction of CHA0. We observed that the lysis and assumedly the production of R-tailocins and more specifically tailocin #1 had a pronounced effect on the survival of the kin competitor Pf-5. Representative

confrontations are shown in Fig. 6 and Supplementary Movie 6, 7, 8. The time-lapse observations show that following the lysis of one CHA0 wild-type cell, on average 17.6 Pf-5 cells were killed (Fig. 6a, Supplementary Table 6, Supplementary Movie 6). Moreover, following the lysis of one CHA0 Δtail2ΔmyoΔsiph cell (producing exclusively tailocin #1), on average 12.6 Pf-5 cells were killed (Fig. 6b, Supplementary Table 6, Supplementary Movie 7). On the contrary, the lysis of CHA0 cells producing only tailocin #2 (CHA0 Δtail1ΔmyoΔsipho) did not affect the survival of Pf-5 cells (Fig. 6c, Supplementary Movie 8). The killing effect of the CHA0 tailocin #1 during kin bacterial competition against Pf-5 was amplified by inducing CHA0 cells with MMC prior to competition (Supplementary Fig. 4, Supplementary Movies 9, 10, 11). Furthermore, we observed that the tailocin #1 of CHA0 could target Pf-5 cells that were not in close vicinity of a tailocin-producing CHA0 cell (Supplementary Movie 12).

Subsequently, we determined to which extent R-tailocins influence the fate of CHA0 populations following 24 h competition with Pf-5 in a classical co-culture assay in liquid and on solid media. The fitness of CHA0 was significantly higher in liquid competitions when it could produce only tailocin #1 instead of only tailocin #2 (Fig. 7). However, this gain in fitness was limited as it did not allow CHA0 to outcompete Pf-5 (Fig. 7a). Conversely, this fitness gain is dependent on environmental conditions as no differences were observed on solid medium (Fig. 7b). We hypothesize that this difference may be due to limited spatial diffusion of R-tailocins on solid medium compared to liquid medium.

This demonstrates that R-tailocins are a small-scale, specialized and targeted weapon that can have a drastic impact on interactions between kin bacteria at micro-scale levels, i.e., in the immediate microenvironment of microbial competitors, which for instance may be relevant when warding off competitors invading discrete locations in biofilms. Compared to investigations by microscopy as highlighted in the present study, classical co-culture assays may not always be an adequate and sensitive tool to detect phage-like particle-mediated microscale interactions.

## Discussion

Bacteria need to be highly competitive to colonize the desired ecological niche to gain access to nutrients and space. The Pp and Pc subgroup bacteria studied here are known to rely on large-spectrum antimicrobials to target genetically disparate microbial competitors[7]. Close relatives are, like the producers, generally insensitive to these toxic molecules[43–45]. However, bacteria are not only in competition with distant or non-related micro-organisms but also with their own kin since phylogenetically closely-related bacteria share most of their genetic content and likely compete for the same ecological niches. Competition against kin bacteria requires highly specific mechanisms as to outcompete kin but not kill individuals from the same clonal population (i.e., strains that have identical genomes). This implies that the arsenal used against kin needs to be highly specialized with a narrow activity spectrum. Viral particles, such as bacteriophages, are known to have a narrow activity spectrum in addition to a specific host spectrum[15]. The findings of our present work on plant-beneficial strains of the Pp and Pc subgroups of *Pseudomonas* are in line with previous work supporting that environmentally and medically important pseudomonads use viral particles (and particularly R-tailocins encoded within their genomes as specialized weapons against kin bacteria[23,25,28,29,33]. Kin discrimination mediated by viral particles was recently described between different strains of *P. aeruginosa*[62]. Importantly, our live cell imaging work now provides unprecedented insight into the dynamic of production and release of R-tailocins

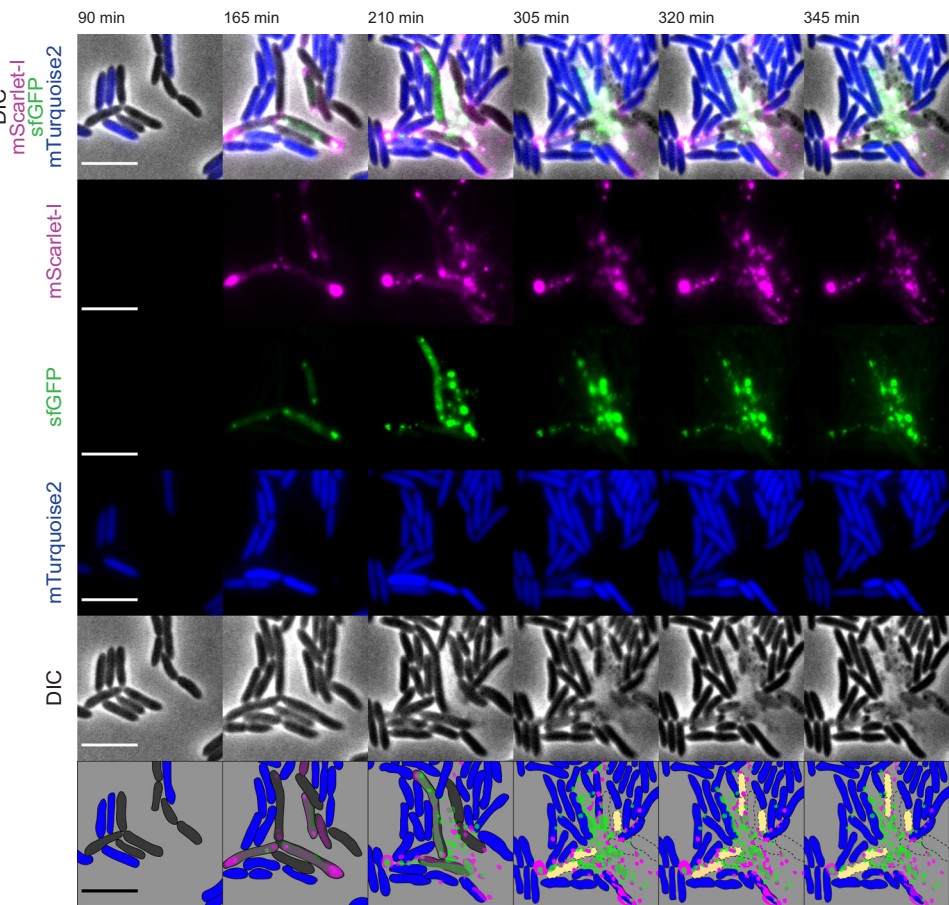

**Fig. 5 Following induction of *Pseudomonas protegens* CHA0 R-tailocins are thrusted in the environment to kill competing *Pseudomonas protegens* Pf-5.** CHA0 cells with tagged tailocin #1 (mScarlet-I) and tailocin #2 (sfGFP) were induced with mitomycin C, washed and confronted with Pf-5 mTurquoise2 cells and monitored with epifluorescence time-lapse microscopy (see Supplementary Movie 4). Cartoons outline the interaction at the different time points. CHA0 and Pf-5 mTurquoise2 live cells are shown in gray and blue, respectively. Ghost cells are outlined with a dotted line and are either not colored (CHA0) or filled with yellow (Pf-5). Tailocins #1 and #2 are depicted in magenta and green, respectively. The scale bar represents 5 μm.

as well as demonstrating their implication in the competition between kin bacteria.

We tagged the tail of the viral particles in C-terminal to be able to monitor the production dynamic of R-tailocins. In most studies that have worked with tagged phages, the capsid or the DNA of these particles were tagged[46–49], however since R-tailocins resemble headless phage particles, we decided to tag the subunit of the tail sheath. Importantly, tagged R-tailocins were expressed from their native chromosomal and thus regulation context. Nonetheless, we found that the labels used had an impact on the overall structure of the tail and inhibited the correct activity of the R-tailocin, presumably by obstructing efficient contraction (Supplementary Fig. 2, Supplementary Table 5). Nevertheless, these fluorescent labels allowed us to visualize the production dynamic of such particles inside induced *P. protegens* CHA0 cells. Surprisingly, the R-tailocins are produced in a sequential manner: originating from the cell center and migrating to the cell poles (Fig. 4, Supplementary Movies 1, 2, 3). This localization could be mediated by various mechanisms that are involved in the polar transport of proteins. Proteins are known to move to cell poles by passive diffusion, by specific affinity to curved membranes or affinity for cell envelope elements or by using gradients such as the one created by the Min system required for cell division site placement[50]. Phage particles have also been shown to travel along tubulin structures for translocation to subcellular compartments[51,52]. Furthermore, it has been reported that during the lytic cycle of the λ phage in *Escherichia coli*, the phage DNA

preferentially locates at the cell poles[46]. Therefore, it is most likely that the polarized transport of R-tailocins is an active mechanism that would allow further propagation of the particles to reach further cells.

The labeling of these viral particles also allowed us to demonstrate that following explosive cell lysis, R-tailocins are thrusted up to several tens of micrometers into the environment where they can reach and kill kin competitor (Figs. 4, 5, Supplementary Movies 3, 4, 5). Explosive cell lysis mediated by an endolysin encoded in the pyocin (tailocin) gene cluster of *P. aeruginosa* has recently been identified as a mechanism for the formation of membrane vesicles and release of eDNA but so far has not been linked to the release of tailocins[53,54]. The turgor pressure in bacterial cells has been estimated up to 1.9 MPa (in comparison a car tire is inflated at ca. 0.2 MPa)[55,56]. This suggests that the explosive disintegration of the bacterial cell wall could liberate sufficient energy to eject the tailocins at the distances observed. Here, we demonstrated that following explosive lysis of cells and viral particle ejection, tailocin #1 is a targeted and efficient microscale weapon against the kin competitor, both in non-induced condition and upon artificial induction by MMC. Indeed, through different time-lapse experiments, we monitored the competition between Pf-5 and various non-induced variants of CHA0 and evidenced the specific role of tailocin #1 of CHA0 against Pf-5 (Fig. 6b, Supplementary Movie 7). We showed that following the release of tailocin #1 from CHA0, closely located Pf-5 cells are killed. We confirmed this hypothesis by artificially

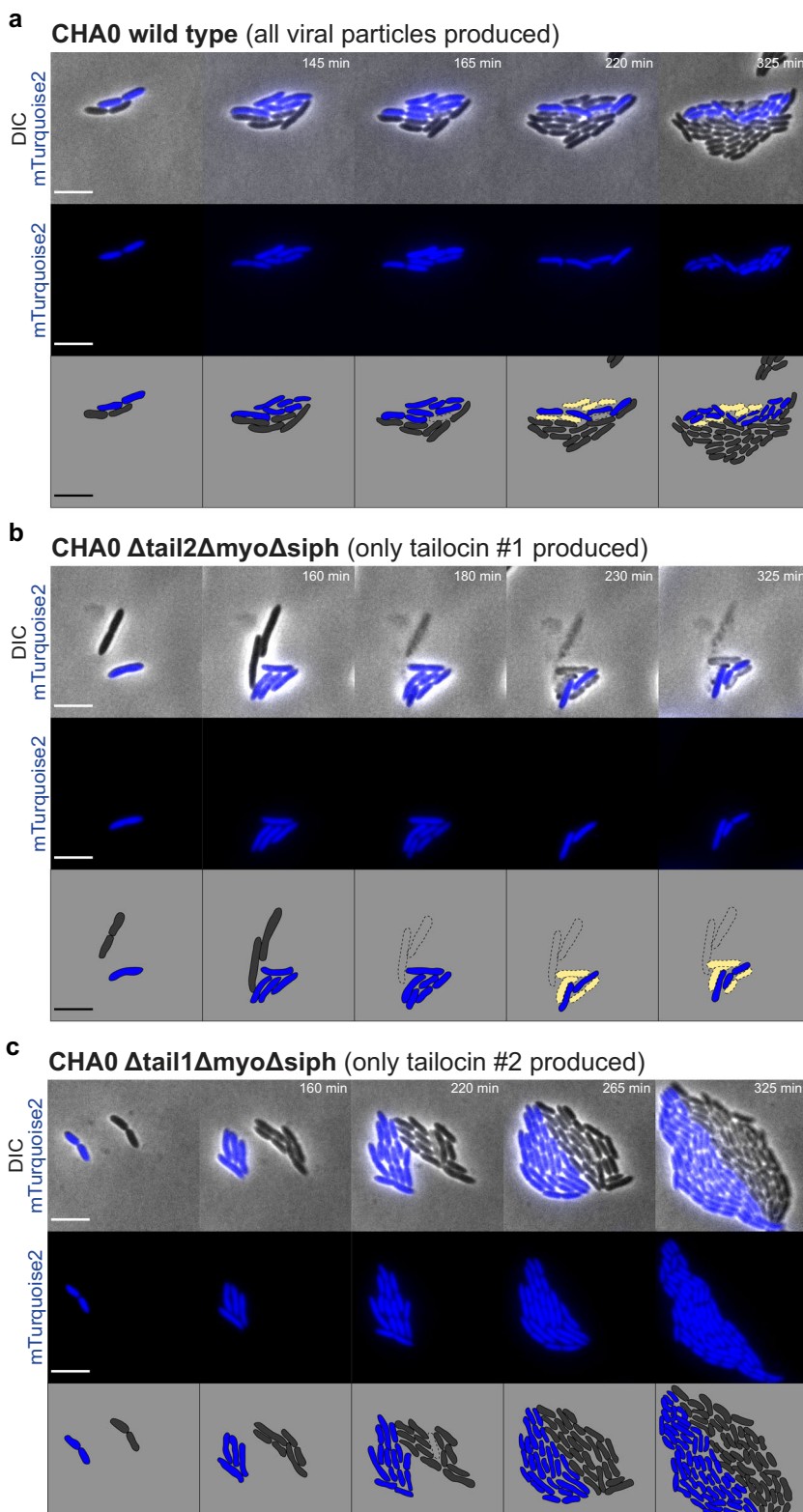

**Fig. 6 Exclusive involvement of tailocin #1 of *Pseudomonas protegens* CHA0 during the direct competition against *Pseudomonas protegens* Pf-5 in non-induced conditions (i.e. no mitomycin C induction of CHA0).** The competition between Pf-5 mTurquoise2 and **a** CHA0 wild type (Supplementary Movie 6), **b** the CHA0 mutant able to produce exclusively tailocin #1 (Δtail2ΔmyoΔsiph) (Supplementary Movie 7), or **c** the CHA0 mutant able to produce only tailocin #2 (Δtail1ΔmyoΔsiph) (Supplementary Movie 8) was followed by time-lapse microscopy. Cartoons outline the interaction at different time points. CHA0 and Pf-5 mTurquoise2 live cells are shown in gray and blue respectively. Ghost cells are outlined with a dotted line and are either not colored (CHA0) or filled with yellow (Pf-5). The scale bar represents 5 μm.

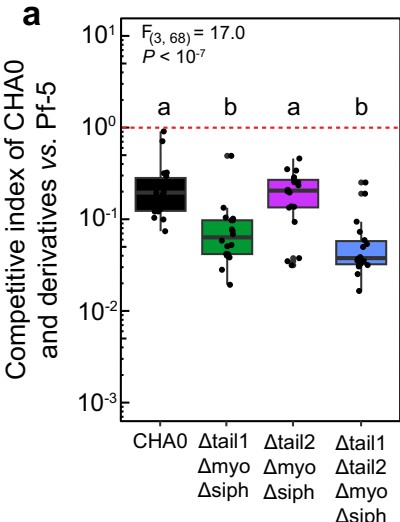
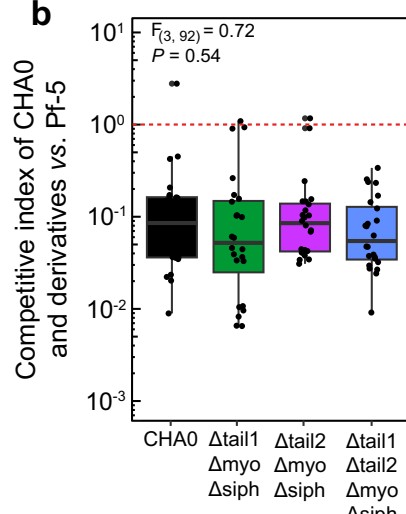

**Fig. 7 Tailocin #1 contributes to the competitiveness of *Pseudomonas protegens* CHA0 toward *P. protegens* Pf-5.** The competitiveness of CHA0 wild type and mutants producing exclusively the tailocin #1 (Δtail2Δmyo Δsiph) or the tailocin #2 (Δtail1Δmyo Δsiph), or none of the viral particles (Δtail1 Δtail2Δmyo Δsiph) was assessed in 1:1 mixtures with Pf-5 during 24 h in liquid (**a**) and on solid (**b**) media. The red dotted line indicates a competition where both strains would not be influenced by the presence of one another. Statistical differences between the competitive indices of CHA0 wild type and mutants in confrontation with Pf-5 were assessed by ANOVA coupled with Tukey's HSD test and are indicated with letters a and b. $n = 8$ biological independent experiments were performed with each 3 technical replicates. The boxes indicate the interquartile range with the center representing the median. The detailed results used for this figure can be found in the Supplementary Data 3.

inducing the production of these viral particles with MMC and observed a greater impact on this sensitive strain (Supplementary Fig. 4b, Supplementary Movie 10). Presently, it is difficult to speculate on possible factors inducing R-tailocin release even in non-artificially induced CHA0 cells during competition with kin competitor Pf-5, but it seems likely that a mechanism of interference competition sensing is involved, such as sensing of DNA damage, membrane perturbation or nutrient shortage caused by the competitor attacks[57,58]. Nevertheless, albeit R-tailocins play a major role in competition at cellular level, their impact is less readily visible following prolonged competition in classical confrontation assays in liquid or on solid media (Fig. 7). We suggest that R-tailocins are more likely an altruistic rather than a spiteful weapon, reducing the invasion from a kin at a microscale level. However, in the micro-environment of a cell deploying its R-tailocin weaponry the consequences for sensitive kin cells are fatal. This may be particularly relevant in a biofilm or micro-colony environment where R-tailocins may serve as specialized and targeted weapons to hold off competitors from invading discrete colonization sites. In contrary to time-lapse microscopy analyses as done here, classical confrontation assays combined with CFU counting seem therefore inapt to fully image the subtle localized effects of the phage-like weaponry of these environmental bacteria.

Although we were able to study in depth the effect of tailocin #1 of CHA0 on Pf-5, we also identified three other viral particle sequences in the genome of CHA0. Interestingly, we found that two out of the four viral particles have very specialized activity spectra affecting either strains from the *P. protegens sensu stricto* species (siphovirus) or strains belonging to the *P. protegens* subgroup (tailocin #1) (Fig. 2a). However, we did not find any targets for the two remaining viral particles. Since tailocin #2 was visualized with TEM (Fig. 2b) and found in all genomes of *Pseudomonas* in this study (Fig. 3b), we hypothesize that this R-tailocin has a different activity spectrum affecting strains that do not belong to the ones tested. R-tailocins targeting phylogenetically distant species that may occupy the same ecological niche have been reported for other environmental bacteria, including

*Pseudomonas putida*[28], *P. fluorescens*[34] and *Xenorhabdus nematophila*[59]. Tailocin #2 of CHA0 should be tested on more distantly-related strains to find targets. Conversely, the myovirus was not detected following viral particle extraction, which explains the inefficiency of extracts towards the selected *Pseudomonas*. We suggest that the myovirus is a cryptic phage, which is a prophage that is no longer able to induce the lysis of its bacterial host and produce phage particles[60]. This is most likely due to genome rearrangements and degeneration of genetic content[61]. We further supported our hypothesis of specialization of viral particles by demonstrating that all viral particles extracted from one strain have more of a lytic effect on other strains belonging to the same subgroup (Fig. 1). As previously mentioned, this may be a mechanism to prevent kin bacteria from invading an already established population, as it was described for *P. aeruginosa* biofilms, in which strains that harbor viral particles had a competitive advantage over those who lacked viral particles[62].

Accordingly, the presence of viral particles, more specifically R-tailocins, is not uncommon in *Pseudomonas* genomes. In the 25 genomes tested in this study, we found a tailocin hotspot region, between *mutS* and *cinA/recA*, which was previously identified as a R-tailocin gene cluster insertion site in some other environmental pseudomonads, including strains of the *P. putida* and *P. fluorescens* groups[28,31]. Within this hotspot, we found that there was at least one R-tailocin, namely tailocin #2 of CHA0 that was conserved among all the *Pseudomonas* tested (Fig. 3). Interestingly, this particular R-tailocin was also identified in genomes of more distantly-related *Pseudomonas*, such as *Pseudomonas corrugata*, *Pseudomonas korensis*, *Pseudomonas putida* and *Pseudomonas syringae*[24,29,31], suggesting that the original prophage giving rise to this specific R-tailocin may have been acquired by a common ancestor of all these *Pseudomonas*. However, genetic differences occurred, specifically for the genes encoding the tail fibers, involved in the cell target recognition. Furthermore, it appears that the other tailocins identified in the selected genomes are specific to each subgroup (Fig. 3b). One of the most simple evolutionary scenarios could be that multiple temperate phages

were independently acquired, leading to different tailocins through evolution. This hypothesis could be supported by the undomesticated prophage present in the genome of *P. chlororaphis* subsp. *aureofaciens* strain O6[19,20,28] that may evolve into a tailocin. Furthermore, as these tailocins may be acquired independently, they have a narrow and specific activity spectrum that may have been obtained through the specificity of the prophage but may also have evolved following acquisition. When compiling all this genomic information, we suggest that prophage domestication by bacteria could be a multi-step evolutionary process[19,20] in conjunction with horizontal gene transfers and recombination between different prophages[22]. Following these steps, the resulting tailocins may be specialized through the genetic modifications of key genes involved in specificity such as the tail fibers, to become precise weapons.

Based on the results of present study, we propose a model where certain R-tailocins are targeted and efficient weapons in microscale competitions between kin bacteria (Fig. 8). We suggest that R-tailocins are induced either spontaneously or by external stressors, including interference competition signals[57,58] or competitor toxins[63]. Following induction, R-tailocins are synthesized at the center of the cell and migrate to the cell poles. Once R-tailocins have migrated, producing cells are lysed in a two-step process, firstly by forming spheroblasts and secondly through complete lysis of the cells. Following explosive lysis, the viral particles are thrusted into the environment where they reach competing bacteria. R-tailocins specifically bind and kill kin bacteria that are ecological niche competitors while exempting more distantly-related strains. This kin-exclusion may allow the reduction of kin bacteria and may favor the input of more phylogenetically distant bacteria that bring new genes involved in different metabolic pathways that would increase the diversity of public goods within the community. Furthermore, clonal cells are not disturbed by the R-tailocins in the environment as bacteria seem to be immune to their own R-tailocins.

To conclude, tailocins and viral particles are likely to participate in the localized "cell-to-cell" competition of bacteria. Learning more about these intriguing viral particles could give us better insight into the complex relationships within bacterial communities. The findings of this study provide a basis for follow-up work into signaling and regulatory mechanisms controlling the induction and formation of these phage tail-like

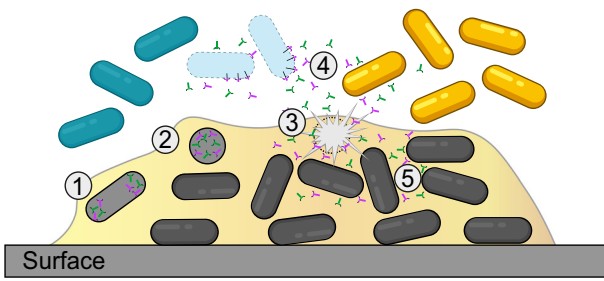

**Fig. 8 Model of the ecological role of R-tailocins in interbacterial competitiveness.** We propose a model where R-tailocins play an important ecological role within a bacterial community such as the ones that compose biofilms. **1** Some cells are induced upon environmental stress and synthesize R-tailocins that are produced at the center of the cell and migrate to the cell poles. **2** Subsequently to the migration of the R-tailocins, the cells lyses, firstly by forming a spheroblast. **3** Secondly, the cell lyses completely and explosively, thereby thrusting its R-tailocins in the environment. **4** Once in the medium, R-tailocins specifically bind to kin bacteria and kill them whereas more distantly-related bacteria are spared. **5** Clonal cells are immune and are therefore protected from the R-tailocins released in the environment.

devices, and molecular determinants that sensitize or shield targeted cells against these bacterial weapons.

## Methods

**Bacterial strains, plasmids and growth conditions.** Strains used in this study are detailed in Supplementary Tables 1 and 2. Bacteria were routinely grown on nutrient agar (NA), in nutrient yeast broth (NYB) or in lysogeny broth (LB) at 25 °C for *Pseudomonas* strains and 37 °C for *Escherichia coli* strains. All liquid cultures were incubated with shaking at 180 rpm. Growth media were supplemented with ampicillin (100 µg mL⁻¹, Amp100), chloramphenicol (50 µg mL⁻¹, Cm50), gentamicin (10 µg mL⁻¹, Gm10) or kanamycin (25 µg mL⁻¹, Km25) when required. All strains were stored at −80 °C in 50 % glycerol [v/v].

**Genomic analysis.** Representative environmental strains of the *P. protegens* (Pp) and *P. chlororaphis* (Pc) subgroups were selected based on their phylogenetic relatedness with *P. protegens* type strain CHA0 and the availability of their genomes (Supplementary Table 1). To build the phylogenetic tree of the selected *Pseudomonas* strains, nucleotide sequences of the housekeeping genes *gyrB*, *rpoB* and *rpoD* were aligned using MUSCLE[64]. The aligned sequences were then concatenated (8289 bp) and used to compute a Maximum Likelihood tree using PhyML in Seaview program v. 4.7[65] with default parameters. The assessment of the tree robustness was done with 100 bootstraps. The obtained tree was drawn with the Interactive Tree Of Life (iTOL) online tool.

The genomes of the selected *Pseudomonas* strains were then analyzed with PHASTER[35] to search for sequences encoding viral particles. For each viral sequence detected, tailocin gene clusters were manually differentiated from prophages by verifying the absence of genes involved in capsid formation. All the proteins encoded by the tailocin gene clusters were then characterized using NCBI Conserved Domain Database Search[66] and Protein Homology/analogY Recognition Engine V 2.0 (Phyre2[67]). The conservation of the tailocin gene clusters amongst *Pseudomonas* was evaluated by inspecting genomic regions located between the *mutS* and *cinA/recA* genes[28]. The synteny and the detection of sequence homology of the clusters was obtained using the MultigeneBlast tool[68] with the homology threshold set at 70 % of identity on at least 70 % of the total gene length.

**Construction of *P. protegens* CHA0 deletion and gene replacement mutants.** CHA0 mutant strains (Supplementary Table 2) were created using the I-SceI system with the suicide vector pEMG adapted to *P. protegens*. Plasmids and primers are listed in Supplementary Tables 7 and 8. The detailed procedures are provided in the Supplementary Information.

**Growth kinetics and MMC sensitivity of CHA0 and its derivatives.** Bacterial growth for CHA0 and its derivatives, with or without the addition of 3 µg mL⁻¹ of MMC, was monitored for 24 h by measuring the OD$_{600nm}$ every 5 min in a BioTeK Synergy H1 plate reader (BioTek Instruments Inc., Winooski, VT, USA). This growth kinetic assay was performed one time with five biological replicates.

**Viral particle extraction and purification.** To extract and purify viral particles from the different *Pseudomonas* strains, cultures were restarted into fresh NYB from an overnight culture. When cultures reached exponential growth phase (OD$_{600nm}$ of 0.4–0.6), 3 µg mL⁻¹ of MMC was added. The cultures were further incubated at 25 °C for a minimum of 3 h and then centrifuged at 6000 rpm for 15 min. Supernatants were filtered with 0.2 µm cellulose acetate filters. To precipitate viral particles, 10% [w/v] of polyethylene glycol 8000 was suspended into the filtered supernatant. Following an overnight incubation at 4 °C with agitation, the mixture was centrifuged at 8000 rpm for 1 h. The supernatant was discarded, and the pellet was suspended in 2 mL of either Tn50 buffer (tailocins) or SM buffer (phages). Following another overnight incubation at 4 °C with agitation, 2 mL of chloroform were added. The mixture was vortexed and then centrifuged at 4000 rpm for 10 min. The aqueous phase, containing the viral particles, was carefully extracted and stored at 4 °C. To extract the individual viral particles from CHA0, mutants encoding only one of the viral particles were used (Supplementary Table 2, 3).

**Visualization of viral particles using transmission electron microscopy.** Two microliters of viral particle suspension were adsorbed on a glow-discharged copper 400 mesh grid coated with carbon (EMS, Hatfield, PA, US) for 1 min at room temperature. Then, they were washed with three drops of distilled water followed by staining with uranyl acetate 1 % in H$_2$O during 1 min. Excess of uranyl acetate was drained on a blotting paper and the grid was dried for 10 min before image acquisition. Micrographs were taken with a Philips CM100 transmission electron microscope (Thermo Fisher Scientific, Hillsboro, USA) at an acceleration voltage of 80 kV with a TemCam-F416 digital camera (TVIPS GmbH, Gauting, Germany).

**Testing bacterial sensitivity to viral particles.** To test the sensitivity of the different bacterial strains, soft agar assays were performed using an adaptation of a

protocol from Hockett and Baltrus[69]. Following an overnight incubation, 500 µL of each bacterial culture was mixed with 20 mL of LB soft agar. This mixture was poured into an empty Petri dish and aliquots of 5 µL of purified viral particles suspension were deposited once the agar solidified. Serial dilutions of the purified viral particles from CHA0 mutants were also spotted. The plates were incubated overnight at 25 °C. The sensitivity of the bacteria toward the viral particles was assessed using the following scale: clear lysis, semi-clear lysis, opaque lysis and no lysis (resistant). The experiment was performed independently twice and was read by three independent people and purified particle extracts from non-induced cultures were spotted as controls.

The sensitivity of Pf-5 towards tailocin #1 of CHA0 was evaluated in liquid assays. Pf-5 was restarted from an overnight culture in fresh NYB. The bacterial cells were washed twice in sterile 0.9 % NaCl solution and adjusted at $OD_{600nm}$ of 1 in a glass tube. Purified tailocin #1 (native or mScarlet-I tagged) suspension in Tn50 buffer (200 µl) was then added to the bacterial suspension (around 400 AU of tailocin, based on semi quantification[70]). As a control, the same volume of Tn50 buffer was used. This experiment was done three times independently. The survival of Pf-5 was monitored by CFU counting.

The sensitivity of Pf-5 towards CHA0 was also assessed in liquid assays. Pf-5, CHA0 and derivatives were restarted from overnight cultures in fresh NYB. When cultures reached exponential growth phase, they were washed twice in sterile 0.9 % NaCl solution and adjusted at an $OD_{600nm}$ of 0.1 in a glass tube. CHA0 adjusted cultures were exposed to 3 µg mL$^{-1}$ of MMC for 30 min. The induced cultures were washed twice with sterile 0.9 % NaCl solution and mixed with the adjusted Pf-5 culture. The survival of Pf-5 was monitored by CFU counting for 150 min taking measurements every 30 min. This experiment was performed three times independently.

**Bacterial competition assays**. The competitiveness of CHA0 and its mutants Δtail2ΔmyoΔsiph (tailocin #1 produced), Δtail1ΔmyoΔsiph (tailocin #2 produced) and Δtail1Δtail2ΔmyoΔsiph (no viral particles produced) (Supplementary Table 2, 3) was assessed in confrontation assays with Pf-5 on solid media and in liquid culture. CHA0 and mutants were marked with a constitutively expressed GFP-tag using pBK-miniTn7-gfp2 (Gm$^R$) to allow distinction between both competing strains. Both assays were performed up to eight times independently.

For competitions on solid media, cells harvested from bacterial overnight cultures were washed with sterile 0.9 % NaCl solution. Cell suspensions were adjusted to an optical density at $OD_{600nm}$ of 0.1. Aliquots of 100 µl of the competition mix, composed out of a 1:1 ratio of each strain, were spotted onto cellulose acetate filters (0.2 µm pore size, Sartorius) placed onto NA plates and incubated at 25 °C for 24 h. The filters were placed into 15 mL falcon tubes containing 5 mL of sterile 0.9 % NaCl solution and tubes vigorously shaken to suspend the bacterial growth. Tenfold serial dilutions were performed and 10 µL aliquots were spotted onto NA plates at t = 0 and NA plates and NA + Gm$_{10}$ plates at t = 24 h. The CFU counts of the competitors (CHA0 or derivatives) and Pf-5 were determined and used to calculate the competitive index (CI) as follows: CI = [CFU$_{Pf-5-24 h}$/CFU$_{CHA0-24 h}$] / [CFU$_{Pf-5-0 h}$/CFU$_{CHA0-0 h}$]. The fluorescence of colonies was visualized under blue light using a Fusion FX Spectra imaging platform (Vilber-Lourmat®).

For competition in liquid media, overnight cultures of CHA0 derivatives tagged with GFP and the bacterial competitor Pf-5 were restarted at 1:100 [v/v] into fresh NYB. Once bacterial cells reached exponential growth phase, all the bacterial cells were washed twice with fresh NYB and adjusted to an $OD_{600nm}$ of 0.1. Bacteria were mixed in glass tubes in 1:1 ratio in a total of 3 mL of NYB and placed at 25 °C without shaking. The survival of CHA0 and derivatives was monitored at t = 0 and t = 24 h by CFU counting using gentamicin as the selection factor. The survival of the competitor strain Pf-5 was monitored at t = 0 and t = 24 h on NA plates, by counting the number of non-GFP colonies.

**Live cell imaging**. For time-lapse experiments, epifluorescence microscopy was executed on a Nikon Inverted Microscope EclipseTi-E, fitted with a Perfect Focus System (PFS), a pE-100 Cool-LED light source and a Plan Apo l 100 ×1.45 Oil objective (Nikon), placed in a room where the temperature is fixed at 22 °C. Light intensity was set at 10 % (Solar light engine, LED power 4 %) and images were captured with an ORCA-flash 4.0 Camera (Hamamatsu). The exposure time for phase contrast was 20 ms, the one for mScarlet-I was either 500 ms or 300 ms, the one for sfGFP and mTurquoise2 was 100 ms. Positions on the patches were set with MicroManager (version 1.4.22). Images were analyzed using Fiji 1.52i[71].

To monitor the production dynamics of the R-tailocins of CHA0 by microscopy, the tail sheath protein of each R-tailocin was tagged with two distinguishable fluorescent markers, i.e. mScarlet-I (tailocin #1) and sfGFP (tailocin #2). Constructions were done using the gene replacement method based on suicide vector pEMG described above to fuse the fluorescent proteins to the 3' ends of the respective sheath proteins via a linker (mScarlet-I: RGSGGEAAAKA; sfGFP: AGAGAG). The tagged sheath proteins TailSheath1-mScarlet and TailSheath2-sfGFP were expressed from the native chromosomal context in the resulting P. protegens strains. To induce the production of fluorescently tagged R-tailocins, 3 µg mL$^{-1}$ of MMC was added to NYB-grown strains, once the culture reached an $OD_{600nm}$ of 0.4–0.6. Once induced, two 5 µL drops of culture were deposited on an agarose patch containing 1/3 NYB and 2/3 H$_2$O, 1 X minimal medium salts, 1 %

agarose and 9 µg mL$^{-1}$ of MMC which was placed on a microscope slide. Prior to starting the time-lapse, the slide and cover-slip were sealed together using melted parafilm. Cells were followed for 4 h, capturing photos every 2 min.

To calculate the proportion of R-tailocin producing cells subsequent to addition of MMC or not in CHA0 (tailocin #1 tagged with mScarlet-I, tailocin #2 tagged with sfGFP), Δtail1ΔmyoΔsiph (tailocin #2 tagged with sfGFP), and Δtail2ΔmyoΔsiph (tailocin #1 tagged with sfGFP), cells at exponential growth stage were induced or not with 3 µg mL$^{-1}$ of MMC. Snapshots from time-lapses were captured either at 3 or 4 h after induction. These were analyzed by SuperSegger[72] using the 100XEc segmentation constant. Induction was assessed by looking at the sfGFP mean fluorescence. Ten positions were analyzed for each CHA0 derivative.

To survey the effect of CHA0 R-tailocins in direct competition with Pf-5 (sensitive strain, tagged with the fluorescent marker mTurquoise2), cultures were restarted into fresh NYB from an overnight culture. For experiments where CHA0 derivatives were induced, cultures were restarted 30 min prior to Pf-5. When CHA0 cultures reached exponential growth phase ($OD_{600nm}$ of 0.4–0.6), 3 µg mL$^{-1}$ of MMC was added and the cultures were incubated for 30 min at 25 °C. CHA0 cultures were washed twice with sterile 0.9 % NaCl solution to remove all MMC and then put in competition with Pf-5. For experiments without MMC induction, CHA0 and Pf-5 were placed in 1:1 ratio competitions once the cultures reached exponential growth phase. Six microliters of the competition mixes were deposited onto 1 % agarose patches supplemented with 1 X minimal medium salts, 4 mM sucrose and 20 µM FeCl$_3$. Agarose patches were placed upside down into a sealed microscopy chamber (Perfusion Chamber, H. Saur Laborbedarf, Germany) containing a maximum of four patches per chamber. A maximum of five positions per patch were recorded. Cells were followed during 8 h and images were captured every 5 min. Cartoons were drawn manually in Adobe Illustrator (v.2020) using the original time lapse figures as templates.

**Statistics and reproducibility**. All experiments were performed at least using three biological replicates as detailed in the figure legends. Data were analyzed using R studio version 3.6.1 and considered significantly different when $P < 0.05$. The data were verified for normal distribution and variance homogeneity. Then t-tests and ANOVA coupled with Fisher's LSD including Bonferroni correction were employed to assess significant differences between conditions.

**Reporting summary**. Further information on research design is available in the Nature Research Reporting Summary linked to this article.

## Data availability
The datasets that support the findings of this study are available from the corresponding author upon reasonable request. The genome sequence of P. protegens Pf1 is available on the ENA/EBI database with the accession number ERS3935804. Source data underlying plots shown in figures are provided in Supplementary Data 1, 2, 3.

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

## Acknowledgements

We thank Roxane Moritz for help with time-lapse microscopy and Aurélie Marchet for help with the analysis of the bacterial lawn assays. We thank Jean Daraspe (Electron Microscopy Facility of the University of Lausanne) for help with the transmission electron microscope. We thank Sara Mitri, Stephan Gruber, Jan-Willem Veening, Jan-Roelof van der Meer and Stefan Kunz (University of Lausanne) for helpful discussions. We are grateful to Laure Weisskopf (University of Fribourg), Jutta Wiese (GEOMAR Helmholtz Centre for Ocean Research Kiel), and Gary B. Huffnagle (University of Michigan Medical School), and Monika Maurhofer (ETH Zurich) for contributing *Pseudomonas* strains. This work was supported by the University of Lausanne, the Swiss National Science Foundation (grant 310030_184666) and by the National Centre in Competence Research in *Microbiomes*.

## Author contributions

J.V., C.M.H. and C.K. conceived and designed the experiments, C.M.H. and J.V. performed the experiments, C.M.H., J.V. and C.K. wrote the manuscript. All authors contributed to the discussion and approved the final manuscript.

## Competing interests

The authors declare no competing interests.
