## [Peer Review File · Communications Biology]

Reviewers' comments:

Reviewer #1 (Remarks to the Author):

Vacheron et al. described production of phage and two phage-derived tailocin particles in *Pseudomonas protogenes* CHA0 strain. Live cell imaging showed that lysis of CHA0 cells producing tailocin-1 leads subsequently to the death of the neighboring cells of the close Pf-5 strain. Authors conclude that tailocins are used as weapon at micro-scale level to compete kin bacteria. To my opinion, the data presented is of potential interest to microbiology community.

While the manuscript is overall clearly written and the experiment performed may support the conclusions, some key control experiments are missing and some text changes are suggested (see below).

Major comments

1. As it was shown (REF 30) that two tailocins can share same lysis module or that tailocin module may control the production of the phage (Argov et al, 2019), and taking into account that two tailocins module have two independent lysin genes, but only one holin and spanin gene (Supl Table 4), efficient lysis and/or production of all 4 viral particles in the CHA0 mutants should be examined. For example, this may explain why myovirus particles were not detected in the mutant.
2. The frequency of the lysis events in CHA0 strain and its derivatives (producing given viral particle or no viral particles) should be estimated from live imaging at all conditions: no treatment and mitomycin treatment. This data is essential for understanding how broad the phenomenon is, and cannot be estimated from the presented images. In addition, in Ref 53 photo-induction of viral particles by fluorescence imaging was described. Thus, condition referred by authors as "native" may represent photo-induced cells. Control for native conditions without fluorescence imaging is required.
3. Supl Fig1 and line 139: This analysis is not informative, as very limited and not-random collection of strains was tested.
4. Line 150: please temper the generalization. The data suggest that SOME viral particles target closely-related strains
5. Fig 1 and Fig 2 : please provide uninduced (no mitomycin C) control
6. Fig 2A: it will be nice to see dose-dependent response for active viral particles, as demonstrated for many other bacteriocins.
7. Fig 2B: strains used for TEM are not described neither in the Method section, nor in the figure legend.
8. Line 179: Genome excision of all particles can be easily examined by PCR, especially of myovirus in wild type CHA0 and in "only myo" mutant. This will provide an insight into the nature of the myovirus.
9. Line 184: "Tailocin diversity is linked to the phylogenetic distance" - this conclusion is too strong: the only obvious difference in Fig 3 is in tailocin #1 (purple vs orange in Pp and Ps, respectively). Other differences require some quantitative analysis.
10. Fig 4 and 5. This is a set of experiments that raises a major concern: what is the scientific value of visualization of inactive protein? Tailocin #1-scarlet was shown by the authors to be inactive, Tailocin#2-gfp can not be tested for activity as no susceptible strain was detected... In any case, unfused Scarlet and GFP should be visualized for control, and the presence of only fused tagged protein should be examined by Western to ensure no free Scarlet or GFP are formed by spontaneous cleavage under experimental conditions.
11. Fig Supl 3 : the data presented in Supl Fig 3 is very convincing. I would suggest moving it to the main text. However:

- a) The experiment presented is not described in the methods or in the main text. What are the conditions of both strains and their ratio at $t=0$? Are Pf-5 cells stationary or exponentially growing? This can make a difference as “tailocin-persistent cells are more prevalent in stationary phase” (Kandel et al., 2020).
- b) Line 295: “effect was amplified”- to state that quantitation should be performed similarly to the data presented in Table S6.
- c) Are all CHA0 mutant strains grow similarly under native and inducing conditions?
- d) In general, I would suggest to present the data in Fig 6 (“native” or photo-induced conditions) and in Supl Fig 3 (mitomycin induced conditions) together and in a same line, including quantitation of lysis events, quantitation of killing events per lysed CHA0 cell and killing of Pf-5 cells in culture with appropriate controls.
12. Fig 7 and lines 296-301. The experiment is not well described in the text. Please provide more details to understand the experiment performed and the results observed. As far as I understand, the authors do not expect difference between wild type and “only tailocin#1” strain, as both of them produce the killer tailocin#1. Do they mean tailocin #2 in line 302? As this experiment is central to evaluate the biological impact of the study and as mentioned by the authors the data presented is problematic (line 300 and 372), I would suggest the authors to perform more experiments to strengthen their conclusions: use “no viral particles” mutant as a control in competition experiments, evaluate the effect of different starting ratios on the outcome of the competition, perform experiment for shorter time, start from exponentially growing bacteria. Of course, no difference in the fitness of different CHA0 strains, as well as no effect of harboring the selection plasmid should be demonstrated.
13. Line 353: Cell lysis by pyocin was identified AND VISUALIZED. Authors may discuss similarities or differences in the observed dynamics of the lysis in these systems.
14. Line 355: cite also McFarland et al, 2015
15. Line 357: “sufficient energy”- did authors base this claim on calculation? If so, it should be provided or referenced.
16. Authors may include in discussion known cases of kin competition and the use of bacteriocins in such competition by other species. For example, Wall 2016 (Annual Review of Microbiology) and references therein.
- Minor comments
17. Supl. Table 5. Tailocin#2 tagged should be GFP, not Scarlett
18. Line 110: “document” should be “documented”

Reviewer #2 (Remarks to the Author):

Vacheron et al. studied the production and release of pyocins and their activity among Pseudomonads. This study is very nice showing tailocins are a kind of projectile weapon and are not just lysing out of the cells, that may give competitive advantages to the cells.

1. The data showing that tailocins migrate to the cell pole and accumulate there is exciting. Is there any data using native pyocins that support the data, such as TEM image of ultrathin sections? Otherwise, the authors may add some discussion of possible consequence of the fluorescent fusion on the protein localization.
2. Fig.4C. It seems that the number of tailocins released have increased between 142 min and 184 min, though cell did not lyse during this time period. Could the authors clear this?

3. Line 246, the authors claim that the tailocins are thrust several tens of micrometers. This conclusion is one of the key parts in this study but looks as if it is made from one data shown. Although the condition can largely influence the spacial distance of tailocin released, is it possible to quantitatively or qualitatively show this with other replicates?
4. Fig. 6 a and b is there any explanation of why some of the Pf-5 cells closer to the lysing CHA0 cells are alive while the ones behind are dead.
5. Related to Fig. 5 and 6, the authors are showing images where Pf-5 and CHA0 are in close contact. Could the authors observe killing by tailocins where these two are not in contact. This data may support better that the tailocins are thrust from the cell and works in a distance.

Minor points

1. Please add explanation of how the cartoons in Fig. 4, 5 and 6 were constructed.

Reviewer #3 (Remarks to the Author):

The manuscript "Live cell dynamics of production, explosive release and killing activity of phage tail-like weapons for *Pseudomonas* kin exclusion" describes the characterization of two *P. protegens* HMW bacteriocins (from the same strain) and describes some interesting visualization studies of the production and release of these particles.

Some of the experiments are interesting, particularly the microscopy work, but it is presented in a way that suggests that little is known about these entities and that the data answers broad questions. For example lines 26-30 of the abstract are areas that have been studied for decades, by many labs, starting with a tremendous amount of work by a Japanese group on the 1960s. Much is known about the mechanism of contractile R-type bacteriocins. Perhaps less is known about the ecological role, but the data here does not shed any light on that subject. Lines 323-325 also make a claim that the work "establishes" what has been known for 50+ years.

The first set of experiments, lines 117-151 is a survey of the killing activity of the products of various strains against other strains. This work has limited value since it was conducted mainly with a spot assay, which can be unreliable, and by the fact that most *Pseudomonads* produce multiple entities with different killing activities. Indeed the authors actually confirm this by the need to do the experiments described in lines 153-182. They made mutations that dissect that production of different bacteriocins allowing them to confirm the biological activities. With this set of data the authors successfully characterized two new HMW bacteriocins. There is however, already quite a long list of known functional R-type bacteriocins, especially among *Pseudomonas* and its relatives.

The bioinformatics section, 184-227, involves an analysis of the different prophage/putative bacteriocins among other *Pseudomonads*. This work could be shortened to a paragraph or two and presented in the supplemental data. This is the type of study most phage biologists do routinely to look for new prophages etc. One can only speculate on the results until, like they did for one strain, experiments are performed.

I am most enthusiastic about the visualization of the production and release of the bacteriocins. This is interesting and new. The authors might consider revising the manuscript to focus just on this data. The experiments visualizing killing is also interesting, but it only shows that R-type bacteriocins can

indeed kill closely related strains, this has been known for some time.

Some minor things

94-96. The killing mechanism of R-type bacteriocins has been shown to be a dissipation of the membrane potential. There isn't really any evidence of membrane "breakage". It's probably much more subtle.

198-200

It's hard to make evolutionary statements about these entities. There has been millions of years of horizontal gene transfer and co-evolution between phages and related bacteriocins. Not all R-type bacteriocins necessarily evolved directly from prophages. Similar with lines 212-213.

218-221 and 426-428 discuss "cargo" protein. This is speculation that would need experimental confirmation.

323-325 R-type bacteriocins are not "headless bacteriophages", they have a specialized function that may have actually predated tailed phages. Experiments to remove the heads of tailed bacteriophages have resulted in particles with either no bactericidal activity or relatively weak bactericidal activity.

Lines 373-375 I'm not sure "defensive" or "offensive" have much meaning in microbial ecology.

Point-by-Point response to the referees' comments

Reviewer #1 (Remarks to the Author):

Vacheron et al. described production of phage and two phage-derived tailocin particles in *Pseudomonas* proteogenes CHA0 strain. Live cell imaging showed that lysis of CHA0 cells producing tailocin-1 leads subsequently to the death of the neighboring cells of the close Pf-5 strain. Authors conclude that tailocins are used as weapon at micro-scale level to compete kin bacteria. To my opinion, the data presented is of potential interest to microbiology community.

While the manuscript is overall clearly written and the experiment performed may support the conclusions, some key control experiments are missing and some text changes are suggested (see below).

Major comments:

1. As it was shown (REF 30) that two tailocins can share same lysis module or that tailocin module may control the production of the phage (Argov et al, 2019), and taking into account that two tailocins module have two independent lysis genes, but only one holin and spanin gene (Supl Table 4), efficient lysis and/or production of all 4 viral particles in the CHA0 mutants should be examined. For example, this may explain why myovirus particles were not detected in the mutant.

>> We have taken up the point raised by the Reviewer #1 and have performed additional experiments. To better understand the effect on lysis, we followed the bacterial growth of CHA0 and its derivatives producing individual viral particles with or without the addition of 3 µg mL⁻¹ of mitomycin C (MMC). The growth was monitored for 24 h by measuring the OD at 600 nm every 5 min. This growth kinetic assay was performed once with five biological replicates. The set-up of the additional experiment is described on lines 503-507 of the revised manuscript.

As shown in the new Supplementary Figures 1a and 1b and now described on lines 178-189 of the revised manuscript, the strain harboring only the cluster encoding the myovirus (Δ tailcluster Δ siph) appears to be resistant to MMC as the growth curve for this mutant continues to increase like for the control strain that no longer has any viral particles (Δ tailcluster Δ myo Δ siph). Conversely, the growth curves of the other CHA0 derivatives are drastically affected by the addition of MMC. Furthermore, as shown in Supplementary Figure 1c, 1d, when MMC is added to a culture of CHA0 Δ tailcluster Δ myo Δ siph that no longer harbors genes involved in lysis, the cells are unnaturally elongated due to the lack of self-lysis. When the same experiment is performed on CHA0 Δ tailcluster Δ siph that still possesses the myovirus gene cluster the same phenotype is observed.

All this, in addition to the fact that we did not observe the myovirus using transmission electron microscopy, corroborates that the myovirus is potentially a cryptic phage.

2. The frequency of the lysis events in CHA0 strain and its derivatives (producing given viral particle or no viral particles) should be estimated from live imaging at all conditions: no treatment and mitomycin treatment. This data is essential for understanding how broad the phenomenon is, and cannot be estimated from the presented images.

>> We agree with the Reviewer #1. We performed supplemental microscopy images and analysis to quantify the proportion of cells producing fluorescent-labeled tailocins with or without MMC (approach described on lines 609-616 in the Material and Methods section of the revised manuscript). These results are now available in Supplementary Fig. 3 and Supplementary Data 2 and are described on lines 272-273 of the revised manuscript.

According to our time-lapse observations, the frequency of induced cells (i.e. that will lyse) of a strain of CHA0 that produces both tagged tailocins, induced with 3 µg/ml of MMC is around 80 % whereas without induction it is around 0.5 %. These induction frequencies are the same despite the genetic background tested (i.e. CHA0 producing only the tailocin #1 or CHA0 producing only the tailocin #2).

In addition, in Ref 53 photo-induction of viral particles by fluorescence imaging was described. Thus, condition referred by authors as native may represent photo-induced cells. Control for native conditions without fluorescence imaging is required.

>> According to the Ref 53, phototoxicity induced around 2.5 cells every 15 min, while in the absence of fluorescence 0.5 cells were induced every 15 minutes in those experiments. Furthermore, there is no information available about the time of exposure in Ref 53. Here, we observed only 0.5% (i.e. on average 2,7 cells induced in a population of 565 cells on average of induced cells following a 4-h-long time-lapse where imaging and thus fluorescence was active every 5 min. We strongly suggest that photoinduction is a minor effect probably due to the very short time of exposure (sfGFP: 100ms; mTurquoise: 100 ms; mScarlet-I: 400 ms). However, following the comment from the Reviewer #1 and for clarity, we changed the word “native” with the word “non-induced” throughout the revised manuscript.

3. Supl Fig1 and line 139: This analysis is not informative, as very limited and not-random collection of strains was tested.

>> We agree with the concern raised by the Reviewer #1 and removed the original Supplementary Fig. 1 and the corresponding line.

4. Line 150: please temper the generalization. The data suggest that SOME viral particles target closely-related strains

>> We agree with the Reviewer #1 and tempered our sentence by changing it to “*Together these results support the hypothesis that **some** viral particles are produced to target **particularly** phylogenetically closely-related strains*”, on lines 147-148 in the revised manuscript.

5. Fig 1 and Fig 2 : please provide uninduced (no mitomycin C) control

>> Following the request of the Reviewer #1, we performed the non-induced control (no MMC induction) using the different mutants of CHA0 as these strains are the main interest of our study (line 542-543 added to the Material and Methods section of the revised manuscript). We tested this control extract from non-induced CHA0 cultures on all the Pseudomonas strains used in this study. No effect was observed and we added these results in the Fig. 2a (Non induced extract).

6. Fig 2A: it will be nice to see dose-dependent response for active viral particles, as demonstrated for many other bacteriocins.

>> Following the request of the Reviewer #1, we performed serial dilutions of the different viral particles from the different CHA0 mutants and spotted them onto bacterial lawns of cells (lines 537-538 added to the Material and Methods section of the revised manuscript). The data are now available in Supplementary Data 1. Furthermore, these results are now mentioned on lines 168 to 170 of the revised manuscript.

7. Fig 2B: strains used for TEM are not described neither in the Method section, nor in the figure legend.

>> We agree with the Reviewer #1. We have now added detailed information in the legend of Fig. 2 of the revised manuscript. Strains used for the preparation of individual viral particles for TEM correspond to CHA5302 ($\Delta tail2\Delta myo\Delta siph$) for the tailocin #1, CHA5301 ($\Delta tail1\Delta myo\Delta siph$) for the tailocin #2, CHA5289 ($\Delta tailcluster\Delta myo$) for the siphovirus, and CHA5292 ($\Delta tailcluster\Delta siph$) for the myovirus.

8. Line 179: Genome excision of all particles can be easily examined by PCR, especially of myovirus in wild type CHA0 and in only myo mutant. This will provide an insight into the nature of the myovirus.

>> We agree with the Reviewer #1 that it could be interesting to investigate deeper in understanding the role of the myovirus in CHA0. However, the myovirus is not at the center of this study. As outlined above in our reply to comment 1 of Reviewer #1, at this point, results from our additional experiments lead us to conclude that the lysis cassette of this prophage is not functional as the induced cells did not burst following MMC induction (Supplementary Fig. 1). Moreover, we can also conclude that either this prophage cannot benefit from the lysis cassettes of the other viral particles harbored by CHA0 as this myovirus was not detected under

TEM or it benefits from the lytic enzymes of the other viral particles but is produced at very low number following MMC induction.

9. Line 184: Tailocin diversity is linked to the phylogenetic distance - this conclusion is too strong: the only obvious difference in Fig 3 is in tailocin #1 (purple vs orange in Pp and Ps, respectively). Other differences require some quantitative analysis.

>> Although we understand the concern of the Reviewer #1, we specifically selected *Pseudomonas* strains that cover in-depth, up-to-date classification of *Pseudomonas* belonging to the *protegens* and *chlororaphis* subgroups. This allows us to hypothesize and draw an evolutive scenario (among others possible) that encompasses the common ancestor of these two taxonomic subgroups. Nevertheless, we tempered this sentence by writing “Tailocin diversity mirrors the phylogenetic distance between pseudomonads.”, line 194.

10. Fig 4 and 5. This is a set of experiments that raises a major concern: what is the scientific value of visualization of inactive protein? Tailocin #1-scarlet was shown by the authors to be inactive, Tailocin#2-gfp can not be tested for activity as no susceptible strain was detected... In any case, unfused Scarlet and GFP should be visualized for control, and the presence of only fused tagged protein should be examined by Western to ensure no free Scarlet or GFP are formed by spontaneous cleavage under experimental conditions.

>> As shown by Supplementary Fig. 2 and Supplementary Movie 4, the tagged particles are not inactive. However, a decrease in activity is visible and we suggest that this effect is due to the presence of the fluorescent protein bound to the sheath of the tailocin, which would lead to a less efficient contraction of the tail. These microscopy data using the tagged tailocin were included in this study to visualize the production dynamic at cellular level of these viral particles and to produce a bioreporter of the induction process. This loss of efficiency oriented us to perform microscopy competition using untagged tailocins.

Finally, we routinely use bacterial strains that are marked with a constitutively expressed GFP or mScarlett-I and never observe such polarization patterns. Moreover, we never observe this polarization pattern by tagging other proteins of this bacterium (Kupferschmied et al., PLoS Pathogens 2014).

11. Fig Supl 3 : the data presented in Supl Fig 3 is very convincing. I would suggest moving it to the main text.

>> We agree with the Reviewer #1. However, the effect is massive due to prior MMC induction and one major aim of this manuscript was to demonstrate the role of tailocin without this massive induction effect. We are convinced that the non-induced condition (Fig. 6, Supplementary Movie 7, 8, 9) remains the most compelling point of this study.

However:

a) The experiment presented is not described in the methods or in the main text. What are the conditions of both strains and their ratio at t=0? Are Pf-5 cells stationary or exponentially growing? This can make a difference as tailocin-persistent cells are more prevalent in stationary phase (Kandel et al., 2020).

>> The Reviewer #1 was right. This experiment is now described in the Material and Methods section from lines 551-557. “The sensitivity of Pf-5 towards CHA0 was also assessed in liquid assays. Pf-5, CHA0 and derivatives were restarted from overnight cultures in fresh NYB. When cultures reached exponential growth phase, they were washed twice in sterile 0.9 % NaCl solution and adjusted at OD_{600nm} of 0.1 in a glass tube. CHA0 adjusted cultures were exposed to $3 \mu g mL^{-1}$ of MMC for 30 min. The induced cultures were washed twice with sterile 0.9 % NaCl solution and mixed with the adjusted Pf-5 culture. The survival of Pf-5 was monitored by CFU counting for 150 min taking measurements every 30 min.”

b) Line 295: effect was amplified - to state that quantitation should be performed similarly to the data presented in Table S6.

>> We do not believe this is necessary as the results are very visual in the Supplementary Fig. 4. Also, the effect is amplified, as it is now shown in the Supplementary Fig. 3, 80 % of the cells are induced following MMC induction. This percentage leads to a higher number of released tailocins that targets more competitors.

c) Are all CHA0 mutant strains grow similarly under native and inducing conditions?

>> Growth kinetics were performed on the different mutants used in this study (Supplementary Fig. 1). CHAO and its derivatives all grow similarly. Under inducing conditions, the mutants that lack all viral particles (Δ tailcluster Δ myo Δ siph) as well as the mutant that harbors only the genetic cluster encoding the myovirus prophage (Δ tailcluster Δ siph) continues to grow significantly compared to the other mutants that lysed following MMC induction. See also our answer to the first comment of the Reviewer #1.

d) In general, I would suggest to present the data in Fig 6 (native or photo-induced conditions) and in Supl Fig 3 (mitomycin induced conditions) together and in a same line, including quantitation of lysis events, quantitation of killing events per lysed CHAO cell and killing of Pf-5 cells in culture with appropriate controls.

>> As outlined before in our response to comment 11 of the Reviewer #1, we prefer to keep the focus on the impact of such viral particles in a non-induced condition which represents the most compelling point of this study and therefore we would prefer not change the way the figures are presented.

12. Fig 7 and lines 296-301. The experiment is not well described in the text. Please provide more details to understand the experiment performed and the results observed. As far as I understand, the authors do not expect difference between wild type and only tailocin#1 strain, as both of them produce the killer tailocin#1. Do they mean tailocin #2 in line 302? As this experiment is central to evaluate the biological impact of the study and as mentioned by the authors the data presented is problematic (line 300 and 372), I would suggest the authors to perform more experiments to strengthen their conclusion

>> The competition experiments were described in detail in the Material and Methods section on lines 526-551 in the original manuscript (corresponding to lines 551-557 in the revised manuscript). Following the suggestion of the Reviewer #1, we performed up to eight new biological replicates including three technical replicates each time, either in liquid or on solid medium (revised Figure 7 and Supplementary Data 3). This allowed us to detect significant effects of the impact of the tailocin #1 during the competition in liquid condition and no effect on solid medium. These results are now presented and discussed more precisely on lines 310-317 of the revised manuscript.

: use no viral particles mutant as a control in completion experiments

>> Following the suggestion of the Reviewer #1, we now included in the competition assays a new mutant lacking all the structural components for the tailocins as well lacking the siphovirus and myovirus prophages (Δ tail1 Δ tail2 Δ myo Δ siph), but that still harbors the lysis cassette of the tailocin gene cluster (see revised Fig. 7 and revised Supplementary Table 2).

, evaluate the effect of different starting ratios on the outcome of the competition, perform experiment for shorter time,

>> The Reviewer #1 is right and we will consider this for future experiments.

start from exponentially growing bacteria.

>> All bacterial cultures are restarted in a fresh medium and are therefore at exponential growth phase when they were mixed together (see Material and Methods section, line 580-581 of the revised manuscript).

Of course, no difference in the fitness of different CHAO strains, as well as no effect of harboring the selection plasmid should be demonstrated.

>> There is no plasmid in all the strains we used as all mutations are chromosomal deletions (see Supplementary Methods in the revised Supplementary Information). Moreover, no differential growth was observed between the different mutants used in this study (see Supplementary Fig. 1 and our response to comment 11c of the Reviewer #1).

13 & 14. Line 353: Cell lysis by pyocin was identified AND VISUALIZED. Authors may discuss similarities or differences in the observed dynamics of the lysis in these systems.

Line 355: cite also McFarland et al, 2015

>> In their Nature Communications 2016 paper, to which we referred, Turnbull and coauthors examined various mutants of pyocin structural genes of *P. aeruginosa* for defects in eDNA release following explosive lysis and stated that “inactivation of pyocin structural components (pyocin tail, tail fibre or tail sheath), had no effect on MV formation” (i.e. explosive lysis). They also state “A prophage endolysin encoded within the R- and F-pyocin gene cluster is essential for explosive cell lysis”.

Further, recent evidence suggests that the regulation of pyocin production in *P. aeruginosa* may be different in *P. fluorescens* group bacteria (Fernandez et al. J Biotechnol 2020) and likely also in *P. protegens*. Indeed, *P. protegens* CHA0 does not harbor an ortholog of *pvtN* encoding the positive regulator of pyocin production in *P. aeruginosa*. These aspects warrant to be addressed in more detail in future studies. Moreover, the operon *alp* directing the Alp cell lysis pathway system described in McFarland et al. PNAS 2015 that is referred as “Programmed Cell Death”, is absent from the genome of CHA0. For this reason, we refrained from adding this reference.

15. Line 357: sufficient energy - did authors base this claim on calculation? If so, it should be provided or referenced.

>> Following the suggestion of the Reviewer #1, we rephrased this sentence to be more speculative : “*This suggests that the explosive disintegration of the bacterial cell wall could liberate sufficient energy to eject the tailocins at the distances observed*”. (lines 374-376 in the revised manuscript).

16. Authors may include in discussion known cases of kin competition and the use of bacteriocins in such competition by other species. For example, Wall 2016 (Annual Review of Microbiology) and references therein.

>> Following the suggestion of the Reviewer #1, we added a recent example of kin discrimination mediated by R-pyocin in *P. aeruginosa* (Oluyombo et al. 2019, mBio), on lines 342-343 and lines 418-421 in the revised manuscript.

Minor comments

17. Supl. Table 5. Tailocin#2 tagged should be GFP, not Scarlett

>> We thank the Reviewer #1 for detecting this. We corrected it to “Tailocin #2 tagged with sfGFP” (Supplementary Table 5 in the revised Supplementary Information).

18. Line 110: document should be documented

>> Agreed. We corrected it (line 110 in the revised manuscript).

Reviewer #2 (Remarks to the Author):

Vacheron et al. studied the production and release of pyocins and their activity among Pseudomonads. This study is very nice showing tailocins are a kind of projectile weapon and are not just lysing out of the cells, that may give competitive advantages to the cells.

>> We thank the Reviewer #2 for the positive feedback.

1. The data showing that tailocins migrate to the cell pole and accumulate there is exciting. Is there any data using native pyocins that support the data, such as TEM image of ultrathin sections? Otherwise, the authors may add some discussion of possible consequence of the fluorescent fusion on the protein localization.

>> We have shown in previous work on *P. protegens* using protein fusions that there is no polarization (see e.g. Kupferschmied et al. PLoS Pathogens 2014). See also our reply to Reviewer #1, comment #10). We could repeatedly follow the movement of these fluorescent foci from the cell center to the cell poles and are therefore very confident that these are the actual tagged tailocins being transported, and not an artefact due to the tagging. In a yet unpublished study, we have also tagged sheath components of the T6SS of CHA0 and could observe protein movement upon sheath contraction. We fully agree with the Reviewer #2 that it would be very exciting to have ultrathin section TEM pictures further detailing the subcellular localization of the tailocin particles prior to release. We have in fact already started to perform cryo-EM investigations on CHA0 cells and their viral particles (as well as the current covid pandemic related restrictions allowed) and if successful, we aim at presenting this work in a separate follow-up manuscript.

2. Fig.4C. It seems that the number of tailocins released have increased between 142 min and 184 min, though cell did not lyse during this time period. Could the authors clear this?

>> The Fig. 4C is a small section of a large patch containing more cells than shown in the section used for this figure. The increase in tagged tailocins in the background is due to cells out of the frame also producing tailocins. We now added this explanation to the legend of Figure 4 (lines 879-881 of the revised manuscript).

3. Line 246, the authors claim that the tailocins are thrust several tens of micrometers. This conclusion is one of the key parts in this study but looks as if it is made from one data shown. Although the condition can largely influence the spacial distance of tailocin released, is it possible to quantitatively or qualitatively show this with other replicates?

>> This is a good point raised by Reviewer #2. Here, we observed this tailocin dispersion in all time-lapse experiments performed (more than 10 different time-lapse experiments with at least 10 different positions observed). We now added this information on lines 256-257 of the revised manuscript.

4. Fig. 6 a and b is there any explanation of why some of the Pf-5 cells closer to the lysing CHA0 cells are alive while the ones behind are dead.

>> This is a good observation by Reviewer #2. We suggest several different explanations. First, although the time-lapses show a two dimensions perspective, tailocins are projected in three dimensions space, and therefore they may not be equally thrust according to the three dimensions position and the orientation of the cell from the substrate they are attached to. Second, the sufficient amount of tailocins to kill one Pf-5 cell might not be reached according to the previous explanation.

5. Related to Fig. 5 and 6, the authors are showing images where Pf-5 and CHA0 are in close contact. Could the authors observe killing by tailocins where these two are not in contact. This data may support better that the tailocins are thrust from the cell and works in a distance.

>> We agree with the Reviewer #2 and added another supplementary movie showing that the competitor Pf-5 is killed by tailocins just released from physically distant CHA0 (Supplementary Movie 12). We induced cells of the CHA0 mutant that produces only the tailocin #1 ($\Delta tail2\Delta myo\Delta siph$) with MMC. Then we washed them to remove the inducer MMC and confronted them to the Pf-5 strain that is tagged with mTurquoise2. We particularly focused on isolated CHA0 cells that will burst and released only the tailocin #1 as well as surrounding Pf-5 cells. We added this information on lines 307-309.

Minor points

1. Please add explanation of how the cartoons in Fig. 4, 5 and 6 were constructed.

>> These were drawn manually using Adobe Illustrator and the original images as a template. We followed the request of Reviewer #2 and added this information on lines 630-631 in the Material & Methods section. Following this comment, in the former Supplementary Fig. 3 (which now corresponds to new Supplementary Fig. 4 in the revised Supplementary Information) we missed a CHAO cell that survived during the time of observation in the panel B We correct this oversight in the new Supplementary Fig. 4.

Reviewer #3 (Remarks to the Author):

The manuscript Live cell dynamics of production, explosive release and killing activity of phage tail-like weapons for *Pseudomonas* kin exclusion describes the characterization of two *P. protegens* HMW bacteriocins (from the same strain) and describes some interesting visualization studies of the production and release of these particles.

Some of the experiments are interesting, particularly the microscopy work, but it is presented in a way that suggests that little is known about these entities and that the data answers broad questions. For example lines 26-30 of the abstract are areas that have been studied for decades, by many labs, starting with a tremendous amount of work by a Japanese group on the 1960s. Much is known about the mechanism of contractile R-type bacteriocins. Perhaps less is known about the ecological role, but the data here does not shed any light on that subject.

Lines 323-325 also make a claim that the work establishes what has been known for 50+ years.

>> We agree with the Reviewer #3 and are well aware that these phage-tail like entities have been known and studied since their discovery by Francois Jacob 1953-1954 and following the tremendous work of the Japanese researchers. Extensive research has been done on their contractile mechanisms, their activity spectra, and their origin, and less on their ecological role. In our study, it was not our intention to claim work that is known since decades and we regret our arguably not always optimal phrasing. In the revised manuscript, we have now improved our phrasing with respect to the existing knowledge about phage-tail like particles in pseudomonads and revised the corresponding sections in the Abstract (lines 26-28) and the Discussion (lines 338-343) as highlighted by the Reviewer. We believe that our study sheds new light on the dynamics of deployment of these particles. Furthermore, the labeling of tailocins and the live cell visualization of their production dynamics and killing activity are unprecedented results, and we thank the Reviewer #3 for the positive comment about this.

The first set of experiments, lines 117-151 is a survey of the killing activity of the products of various strains against other strains. This work has limited value since it was conducted mainly with a spot assay, which can be unreliable, and by the fact that most *Pseudomonads* produce multiple entities with different killing activities. Indeed the authors actually confirm this by the need to do the experiments described in lines 153-182. They made mutations that dissect that production of different bacteriocins allowing them to confirm the biological activities. With this set of data the authors successfully characterized two new HMW bacteriocins. There is however, already quite a long list of known functional R-type bacteriocins, especially among *Pseudomonas* and its relatives.

>> We are aware that the spot assays can be unreliable, however they allow a quick way of assessing the activity spectra of such viral particles which allowed us to focus on the model competitions between CHA0 and its kin competitor Pf-5. This was a pivotal step finally leading to the visual demonstration of the production, release and targeted activity of tailocin particles, which was at the center of this study. The part of our study dealing with the description of the diversity and intraspecific activity of these particles in a particular subgroup of pseudomonads, i.e. in *P. protegens* and *P. chlororaphis* that are environmental pseudomonads with plant-beneficial activities, per se may not be overly exciting against the knowledge background as discussed above. However, we think that it is important to have this information also in our manuscript because it provides the detailed context on which builds the central live cell imaging work. Also studies providing insight into intraspecific, strain-level diversity and activity of tailocins and into deconstructing viral particle production in a bacterial strain still are very rare and we provide one example for this.

The bioinformatics section, 184-227, involves an analysis of the different prophage/putative bacteriocins among other *Pseudomonads*. This work could be shortened to a paragraph or two and presented in the supplemental data. This is the type of study most phage biologists do routinely to look for new prophages etc. One can only speculate on the results until, like they did for one strain, experiments are performed.

>> Although we understand the concern of the Reviewer #3, we think that it is important to understand the diversity of the viral particles harbored by each bacterial strain as it still brings information on the distribution and diversity of such particles inside bacterial genomes that notably are very closely related and reflect strain-

level intraspecific diversity in the present study. For example, it highlights that up to four different tailocin clusters can be found between this *mutS-recA* region in *Pseudomonas* sp. CMR5c and CMR12a. To our knowledge, a maximum of three tailocins had been reported by Dorosky *et al.*, 2017 and highlighted by Patz *et al.*, 2019, showing the relevance of such bioinformatic analyses.

I am most enthusiastic about the visualization of the production and release of the bacteriocins. This is interesting and new. The authors might consider revising the manuscript to focus just on this data. The experiments visualizing killing is also interesting, but it only shows that R-type bacteriocins can indeed kill closely related strains, this has been known for some time.

>> We agree with the Reviewer #3 that R-type tailocins (and notably R-type pyocins) have been shown to exhibit a small activity spectrum, killing closely related bacteria as well as broader spectrum, however, the visualization at single cell-level, using time-lapse microscopy has never been done so far. Most killing assays have focused on spot assays, which as Reviewer #3 mentioned prior are not always reliable, or CFU counting using competition methods such as liquid assays. As stated above in our response to the second point raised by the Reviewer #3, we are persuaded that the part of our work specifying the intraspecific diversity and activity of the phage tail-like particles in this particular subgroup of environmental pseudomonads and their deconstruction in a model strain is pivotal to our manuscript. It provides the detailed and relevant context on which builds the live cell imaging work that is central to our study.

Some minor things

94-96. The killing mechanism of R-type bacteriocins has been shown to be a dissipation of the membrane potential. There isn't really any evidence of membrane breakage . It's probably much more subtle.

>> We agree with the Reviewer #3, and changed it to "a dissipation of the membrane potential" (lines 94-96 of the revised manuscript).

198-200

It's hard to make evolutionary statements about these entities. There has been millions of years of horizontal gene transfer and co-evolution between phages and related bacteriocins. Not all R-type bacteriocins necessarily evolved directly from prophages. Similar with lines 212-213.

>> We agree with the Reviewer #3 that not each individual tailocin has evolved directly from one unique phage infection. As illustrated in Patz *et al.*, 2019, tailocins can also evolve from duplication events where from one original infection multiple different tailocins can evolve. Furthermore, there can be horizontal gene transfer between different tailocins and phages. In our study, we had described a simplified version of differential and independent phage infections that may lead to the current distribution of the actual tailocins inside *Pseudomonas* genomes. We have now tempered our statement and further discussed the evolutionary origin of these particles on lines 435-446 of the revised manuscript.

218-221 and 426-428 discuss cargo protein. This is speculation that would need experimental confirmation.

>> We agree with the Reviewer #3 that this is a speculation and must be confirmed by experimentation. For clarity, we decided to remove the sentences referring to this speculation. Accordingly, we also changed the designation of the corresponding gene in Fig. 3a to "unknown function" in the revised manuscript.

323-325 R-type bacteriocins are not headless bacteriophages , they have a specialized function that may have actually predated tailed phages. Experiments to remove the heads of tailed bacteriophages have resulted in particles with either no bactericidal activity or relatively weak bactericidal activity.

>> We agree with the Reviewer #3 that "headless bacteriophages" could be perceived as a popularized term to designate R-type tailocins and we change it to "R-tailocins" on line 341 of the revised manuscript.

Lines 373-375 I'm not sure defensive or offensive have much meaning in microbial ecology.

>> We agree with the Reviewer #3 and changed it with the appropriate terms used in ecology according to the publication of Stuart A. West and Andy Gardner, 2010, *Science*, DOI: 10.1126/science.1178332.

We changed the word "defensive" with "altruism" and "offensive" with "spite" on line 391-393 of the revised manuscript.

REVIEWERS' COMMENTS:

Reviewer #1 (Remarks to the Author):

The authors properly responded to all points. No additional comments.

Reviewer #2 (Remarks to the Author):

The authors have cleared my concerns and I do not have further comments.

Reviewer #3 (Remarks to the Author):

No specific comments.

Authors responses

We are grateful for the careful evaluation and for the comments and suggestions made by the Editor and the three Reviewers, which helped us to further improve the quality of our manuscript. We thank the Editor and the three Reviewers very much and we are happy that there are no more concerns or requests following the revision of our manuscript.

REVIEWERS' COMMENTS:

Reviewer #1 (Remarks to the Author):

The authors properly responded to all points. No additional comments.

Reviewer #2 (Remarks to the Author):

The authors have cleared my concerns and I do not have further comments.

Reviewer #3 (Remarks to the Author):

No specific comments.